# AdaWM: Adaptive World Model based Planning for Autonomous Driving

**Hang Wang**[1,2]* **Xin Ye**[1] **Feng Tao**[1] **Chenbin Pan**[1] **Abhirup Mallik**[1]
**Burhaneddin Yaman**[1✉] **Liu Ren**[1] **Junshan Zhang**[2]
[1]Bosch Research North America & Bosch Center for Artificial Intelligence (BCAI)
[2]University of California, Davis
{whang,jazh}@ucdavis.edu
{burhaneddin.yaman,xin.ye3,feng.tao2,chenbin.pan,abhirup.mallik,liu.ren}@us.bosch.com

## Abstract

World model based reinforcement learning (RL) has emerged as a promising approach for autonomous driving, which learns a latent dynamics model and uses it to train a planning policy. To speed up the learning process, the pretrain-finetune paradigm is often used, where online RL is initialized by a pretrained model and a policy learned offline. However, naively performing such initialization in RL may result in dramatic performance degradation during the online interactions in the new task. To tackle this challenge, we first analyze the performance degradation and identify two primary root causes therein: the mismatch of the planning policy and the mismatch of the dynamics model, due to distribution shift. We further analyze the effects of these factors on performance degradation during finetuning, and our findings reveal that the choice of finetuning strategies plays a pivotal role in mitigating these effects. We then introduce AdaWM, an Adaptive World Model based planning method, featuring two key steps: (a) mismatch identification, which quantifies the mismatches and informs the finetuning strategy, and (b) alignment-driven finetuning, which selectively updates either the policy or the model as needed using efficient low-rank updates. Extensive experiments on the challenging CARLA driving tasks demonstrate that AdaWM significantly improves the finetuning process, resulting in more robust and efficient performance in autonomous driving systems.

## 1 Introduction

Automated vehicles (AVs) are poised to revolutionize future mobility systems with enhanced safety and efficiency Yurtsever et al. (2020); Kalra & Paddock (2016); Maurer et al. (2016). Despite significant progress Teng et al. (2023); Hu et al. (2023); Jiang et al. (2023), developing AVs capable of navigating complex, diverse real-world scenarios remains challenging, particularly in unforeseen situations Campbell et al. (2010); Chen et al. (2024). Autonomous vehicles must learn the complex dynamics of environments, predict future scenarios accurately and swiftly, and take timely actions such as emergency braking. Thus motivated, in this work, we devise adaptive world model to advance embodied AI and improve the planning capability of autonomous driving systems.

World model (WM) based reinforcement learning (RL) has emerged as a promising self-supervised approach for autonomous driving Chen et al. (2024); Wang et al. (2024); Guan et al. (2024); Li et al. (2024). This end-to-end method maps sensory inputs directly to control outputs, offering improved efficiency and robustness over traditional modular architectures Yurtsever et al. (2020); Chen et al. (2024). By learning a latent dynamics model from observations and actions, the system can predict future events and optimize policy decisions, enhancing generalization across diverse environments. Recent models like DreamerV2 and DreamerV3 have demonstrated strong performance across both 2D and 3D environments Hafner et al. (2020; 2023).

---

*  Work done while interned at Bosch Research North America.
✉ Corresponding author.

However, learning a world model and a planning policy from scratch can be prohibitively time-consuming, especially in autonomous driving, where the state space is vast and the driving environment the vehicle might encounter can be very complex Ibarz et al. (2021); Kiran et al. (2021). Moreover, the learned model may still perform poorly on unseen scenarios Uchendu et al. (2023); Wexler et al. (2022); Liu et al. (2021). These challenges have led to the adoption of pretraining and finetuning paradigms, which aim to accelerate learning and improve performance Julian et al. (2021). In this approach, models are first pretrained on large, often offline datasets, allowing them to capture general features that apply across various environments. Following pretraining, the model is finetuned using task-specific data to adapt to the new environment. Nevertheless, without a well-crafted finetuning strategy, a pretrained model can suffer *significant performance degradation* due to the distribution shift between pretraining tasks and the new task. To illustrate the possible inefficiencies of some commonly used finetuning strategies, such as alternating between updating the world model in one step and the policy in the next, we present the following motivating example.

**A Motivating Example.** Consider an agent pretrained to make right turns at a four-way intersection, later finetuned for left turns under similar traffic conditions. We evaluate three finetuning strategies: alternate finetuning (Model+Policy), model-only finetuning, and policy-only finetuning. As shown in Figure 1, all strategies initially experience performance degradation due to distribution shift. However, model-only finetuning demonstrates significantly faster recovery compared to the other approaches. This observation reveals a crucial insight: the transition from right to left turns primarily challenges the agent's dynamics model, which must adapt to different spatial-temporal relationships of approaching vehicles. When the dynamics model misaligns with the new environment, the policy inevitably makes decisions based on inaccurate predictions. Thus, dynamics model mismatch becomes the *dominating factor* limiting performance. Model-only finetuning addresses this directly, while alternate and policy-only strategies struggle by failing to prioritize this critical misalignment.

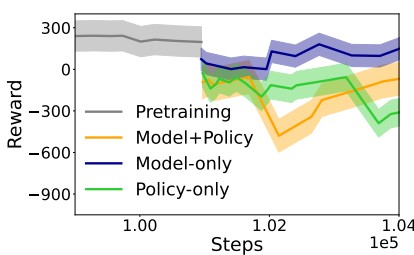

Figure 1: Performance comparison of different finetuning strategies in the left turn with moderate traffic flow task.

This example illustrates that effective finetuning requires identifying and prioritizing the *dominating mismatch* rather than simply alternating between model and policy updates. This work addresses the key challenge of determining efficient finetuning strategies by investigating:

> *When and how should the pretrained dynamics model and planning policy be finetuned to effectively mitigate performance degradation due to distribution shift?*

Building on the insights from our motivated example, we propose AdaWM, an adaptive world model based planning method designed to fully leverage a pretrained policy and world model while addressing performance degradation during finetuning. Specifically, our main contributions can be summarized as follows:

- We quantify the performance gap observed during finetuning and identify two primary root causes: (1) dynamics model mismatch, and (2) policy mismatch. We then assess the corresponding impact of each on the finetuning performance.

- Based on our theoretical analysis, we introduce AdaWM, Adaptive World Model based planning for autonomous driving. As shown in Figure 2, AdaWM achieves effective finetuning through two key steps: (1) *Mismatch Identification*. At each finetuning step, AdaWM first evaluates the degree of distribution shift to determine the dominating mismatch that causes the performance degradation, and (2) *Alignment-driven Finetuning*, which determines to update either the dynamics model or the policy to mitigate performance drop. This selective approach ensures that the more dominating mismatch, as identified in the motivated example, is addressed first. Moreover, AdaWM incorporates efficient update methods for both the dynamics model and the policy, respectively. For dynamics model finetuning, we propose a LoRa-based low-rank adaptation Hu et al. (2021); Koohpayegani et al. (2024), where only the low-dimensional vectors are updated to enable a more efficient finetuning. For policy finetuning, we decompose the policy network

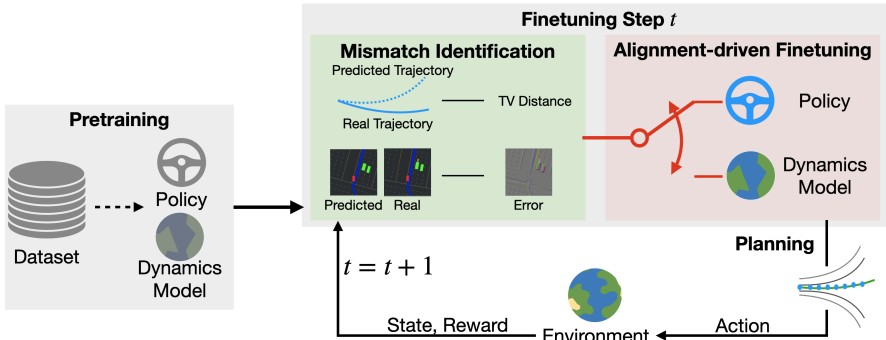

Figure 2: A sketch of adaptive world model based planning (AdaWM): During pretraining, a dynamics model and a planning policy are learned offline. For online adaptation, at each finetuning step $t$, AdaWM first identifies the more dominating mismatch that causes the performance degradation and then carries out alignment-driven finetuning accordingly.

Table 1: Categorization of related works in terms of (1) Learning Method (SL: Supervised Learning; RL: Reinforcement Learning), (2) Finetuning strategy, (3) Online interaction and (4) Tasks.

| Paper | Learning Method | Finetuning | Online | Tasks |
|---|---|---|---|---|
| VAD Jiang et al. (2023) | SL | ✗ | ✗ | CARLA |
| UniAD Hu et al. (2023) | SL | ✗ | ✗ | CARLA |
| Li et al. (2024) | RL | ✗ | ✓ | CARLA |
| Feng et al. (2023) | RL | ✓(Model-only) | ✓ | xArm, D4RL |
| Baker et al. (2022); Hansen et al. (2022) | RL | ✓(Policy-only) | ✓ | Manipulation |
| **AdaWM (Ours)** | RL | ✓(Alignment-driven Finetuning) | ✓ | CARLA |

into a weighted convex ensemble of sub-units and update only the weights of these sub-units to further streamline the finetuning process.

- We validate AdaWM on the challenging CARLA environment over a number of tasks, demonstrating its ability to achieve superior performance in terms of routing success rate (SR) and time-to-collision (TTC). Our results show that AdaWM effectively mitigates performance drops cross various new tasks, confirming the importance of identifying and addressing the dominating mismatch in the finetuning process.

**Related Work.** Recently, end-to-end autonomous driving systems have gained significant momentum with the availability of large-scale datasets Hu et al. (2023); Jiang et al. (2023); Chen et al. (2024). Recent advances in autonomous driving have been driven by two main approaches: supervised learning methods and world model based reinforcement learning methods. Supervised learning approaches like VAD Jiang et al. (2023) and UniAD Hu et al. (2023) have demonstrated impressive performance in the CARLA benchmark. Meanwhile, world model (WM) based methods have emerged as a promising alternative. By learning a differentiable latent dynamics model, world model based methods enable the agent to efficiently anticipate future states and hence improve the decision making Levine & Koltun (2013); Wang et al. (2019); Zhu et al. (2020). However, existing WM-based approaches primarily focus on the pretraining stage, which aims to learn a good policy to solve the new task by using offline datasets. Think2Drive Li et al. (2024) uses DreamerV3 Hafner et al. (2023) for offline training, while DriveDreamer Wang et al. (2023) employs diffusion models for environmental representation. Vasudevan et al. (2024) trains different world models based on different behavior prediction in the environment. While these methods show promise, they often struggle with performance degradation when adapting to new tasks, as shown in our experiments ( i.e., Table 2).

Current studies on finetuning strategies in RL focus on model adaptation Feng et al. (2023) or policy updates through imitation learning Baker et al. (2022); Hansen et al. (2022); Uchendu et al. (2023) and self-supervised learning Ouyang et al. (2022). As summarized in Table 1, existing approaches either lack online fine-tuning capabilities (VAD, UniAD) or employ single-focused strategies (model-only or policy-only). Meanwhile, while Julian et al. (2021); Feng et al. (2023) demonstrated the general effectiveness of finetuning in *robotic learning tasks*, existing autonomous driving

systems lack a unified framework for dynamic adaptation. Our proposed AdaWM addresses these limitations through alignment-driven finetuning that adaptively balances both model and policy updates based on real-time task requirements, achieving superior performance compared to both supervised learning methods without finetuning and RL approaches with rigid adaptation strategies.

## 2 AdaWM: Adaptive World Model based Planning

**Basic Setting.** Without loss of generality, we model the agent's decision making problem as a Markov Decision Process (MDP), defined as $\langle \mathcal{S}, \mathcal{A}, P, r, \gamma \rangle$. In particular, $\mathcal{S} \subseteq \mathbb{R}^{d_s}$ represents state space, and $\mathcal{A}$ is the action space for agent, respectively. $\gamma$ is the discounting factor. At each time step $t$, based on observations $s_t \in \mathcal{S}$, agent takes an action $a_t$ according to a planning policy $\pi : \mathcal{S} \to \mathcal{A}$. The environment then transitions from state $s_t$ to state $s_{t+1}$ following the state transition dynamics $P(s_{t+1}|s_t, a_t) : \mathcal{S} \times \mathcal{A} \times \mathcal{S} \to [0, 1]$. In turn, the agent receives a reward $r_t := r(s_t, a_t)$.

**Pretraining: World Model based RL.** In the pretraining phase, we aim to learn a dynamics model $\text{WM}_\phi$ to capture the latent dynamics of the environment and predict the future environment state. To train a world model, the observation $s_t$ is encoded into a latent state $z_t \in \mathcal{Z} \subseteq \mathbb{R}^d$ using an autoencoder $q_\phi$ Kingma (2013). Meanwhile, the hidden state $h_t$ is incorporated into the encoding process to capture contextual information from current observations Hafner et al. (2023; 2020), i.e., $z_t \sim q_\phi(z_t|h_t, s_t)$, where $h_t$ stores historical information relevant to the current state. We further denote $x_t$ as the model state, defined as follows,

$$\text{Model State } x_t := [h_t, z_t] \in \mathcal{X}, \tag{1}$$

The dynamics model $\text{WM}_\phi$ predicts future states using a recurrent model, such as an Recurrent Neural Network (RNN) Medsker et al. (2001), which forecasts the next hidden state $h_{t+1}$ based on the current action $a_t$ and model state $x_t$.

The obtained hidden state $h_{t+1}$ is then used to predict the next latent state $\hat{z}_{t+1} \sim p_\phi(\cdot|h_{t+1})$. The dynamics model $\text{WM}_\phi(x_{t+1}|x_t, a_t)$ is trained by minimizing the prediction error between these predicted future states and the actual observations. Additionally, the learned (world) model can also predict rewards and episode termination signals by incorporating the reward prediction loss Hafner et al. (2023).

Once trained, the dynamics model is used to guide policy learning. The agent learns a planning policy $\pi_\omega(\cdot|x_t)$ by maximizing its value function

$$V^{\pi_\omega}_{\text{WM}_\phi}(x_t) \triangleq \mathbf{E}_{x_t \sim \text{WM}_\phi, a \sim \pi_\omega}[\sum_{i=1}^{K} \gamma^i r(x_{t+i-1}, a_{t+i-1}) + \gamma^K Q^{\pi_\omega}(x_{t+K}, a_{t+K})], \tag{2}$$

where the first term is the cumulative reward from a $K$-step lookahead using the learned dynamics model, and the second term is the Q-function $Q^{\pi_\omega}(x_t, a_t) = \mathbf{E}_{a \sim \pi_\omega}[\sum_t \gamma^t r(x_t, a_t)]$, which corresponds to the expected cumulative reward when action $a_t$ is taken at state $x_t$, and the policy $\pi_\omega$ is followed thereafter. In this way, the policy learning is informed by predictions from the dynamics model regarding future state transitions and expected rewards.

**Finetuning with the Pretrained Dynamics Model and Policy.** Once the pretraining is complete, the focus is to finetune the pretrained dynamics model $\text{WM}_\phi$ and policy $\pi_\omega$ to adapt to the new task while *minimizing performance degradation* due to the distribution shift. In particular, at each finetuning step $t$, the agent conducts planning using the current policy $\pi_{\omega_t}$ and dynamics model $\text{WM}_{\phi_t}$. By interacting with environments, the agent is able to collect samples $\{x_t, a_t, r_t\}$ for the new task. In this work, the primary objective is to develop an efficient finetuning strategy to mitigate the overall performance degradation during finetuning. To this end, in what follows, we first analyze the performance degradation that occurs during finetuning due to the distribution shift. From this analysis, we identify two root causes contributing to the degradation: mismatch of the dynamics model and mismatch of the policy.

## 2.1 Impact of Mismatches on Performance Degradation

**Performance Gap.** When the pretrained model and policy align well with the state transition dynamics and the new task, the learning performance should remain *consistent*. However, in practice, distribution shifts between the pretraining tasks and the new task can lead to suboptimal planning and degraded performance when directly using the pretrained model and policy. To formalize this, we let $\text{WM}_\phi(P)$ denote the pretrained dynamics model with probability transition matrix $P$, and $\text{WM}_\phi(\hat{P})$ the dynamics model of the new task with probability transition matrix $\hat{P}$. For simplicity, we denote the policy as $\pi = \pi_\omega$ when the context is clear to avoid ambiguity.

Let $\eta$ denote the learning performance of using model $\text{WM}_\phi$ and policy $\pi$ in the pretraining task, and let $\hat{\eta}$ represent the performance of applying $\text{WM}_\phi(P)$ and policy $\pi$ to the new task with environment dynamics $\hat{P}$. Using the value function $V(x)$ defined in Equation (2), the performance gap can be expressed as follows:

$$\eta - \hat{\eta} = \mathbf{E}_{x \sim \rho_0} \left[ V^\pi_{\text{WM}_\phi(P)}(x) - V^\pi_{\text{WM}_\phi(\hat{P})}(x) \right], \tag{3}$$

where $\rho_0$ is the initial state distribution and the second term captures the expected return when the agent applies the pretrained model and policy to the new task with the underlying dynamics $\hat{P}$. Next, we introduce the latent dynamics model to capture temporal dependencies of the environment.

**Latent Dynamics Model.** At each time step $t$, the agent will leverage the dynamics model $\text{WM}_\phi$ to generate imaginary trajectories $\{x_{t+k}, a_{t+k}\}_{k=1}^K$ over a lookahead horizon $K > 0$. These trajectories are generated based on the current model state $x_t$ and actions $a_t$ sampled from policy $\pi$. Particularly, the dynamics model is typically implemented as a RNN, which computes the hidden states $h_t$ and state presentation $z_t$ as follows,

$$h_{t+1} = f_h(x_t, a_t), \quad z_{t+1} = f_z(h_{t+1}),$$

where $f_h$ maps the current state and action to the next hidden state and $f_z$ maps the hidden state to the next state representation. In our theoretical analysis, following the formulation as in previous works Lim et al. (2021); Wu et al. (2021), we choose $f_h = Ax_t + \sigma_h(Wx_t + Ua_t + b)$ and $f_z = \sigma_z(Vh_{t+1})$, where matrices $A, W, U, V, b$ are trainable parameters. Meanwhile, $\sigma_z$ is the $L_z$-Lipschitz activation functions for the state representation and $\sigma_h$ is a $L_h$-Lipschitz element-wise activation function (e.g., ReLU Agarap (2018)). Next, we make the following standard assumptions on latent dynamics model, action and reward functions.

**Assumption 1 (Weight Matrices)** *The Frobenius norms of weight matrices $W$, $U$ and $V$ are upper bounded by $B_W$, $B_U$ and $B_V$, respectively.*

**Assumption 2 (Action and Policy)** *The action input is upper bounded, i.e., $|a_t| \le B_a$, $t = 1, \cdots$. Additionally, the policy $\pi$ is $L_a$-Lipschitz, i.e., for any two states $x, x' \in \mathcal{X}$, we have $d_X(\pi(\cdot|x) - \pi(\cdot|x')) \le L_a d_X(x, x')$, where $d_A$ and $d_X$ are the corresponding distance metrics defined in the action space and state space.*

**Assumption 3** *The reward function $r(x, a)$ is $L_r$-Lipschitz, i.e., for all $x, x' \in \mathcal{X}$ and $a, a' \in \mathcal{A}$, we have $|r(x, a) - r(x, a')| \le L_r(d_X(x, x') + d_A(a, a'))$, where $d_X$ and $d_A$ are the corresponding metrics in the state space and action space, respectively.*

**Characterization of Performance Gap.** We start by analyzing the state prediction error at prediction step $k = 1, 2, \cdots, K$, defined as $\epsilon_k = x_k - \hat{x}_k$, where $x_k$ is the underlying true state representation in the new task and $\hat{x}_k$ is the predicted state representation by using the pretrained dynamics model $\text{WM}_\phi$. The prediction error arises due to a combination of factors such as distribution shift between tasks and the generalization limitations of the pretrained dynamics model. To this end, we decompose the prediction error into two terms,

$$\epsilon_k = (x_k - \bar{x}_k) + (\bar{x}_k - \hat{x}_k) \tag{4}$$

where $\bar{x}_k$ is the underlying true state representation when planning is conducted in the pretraining tasks. The first term $(x_k - \bar{x}_k)$ captures the difference between the true states in the current and pretraining tasks, reflecting the distribution shift between the tasks. The second term $(\bar{x}_k - \hat{x}_k)$

stems from the prediction error of the pretrained RNN model on the pretraining task. This decomposition allows us to rigorously examine the impact of distribution shift and model generalization by bounding these two components respectively.

To analyze the prediction error, we assume the dynamics model is trained using supervised learning on samples of state-action-state sequence and resulting the empirical loss is $l_n$. We assume the expected Total Variation (TV) distance between the true state transition probability $P$ and the predicted dynamics $\hat{P}$ be upper bounded by $\mathcal{E}_{\hat{P}}$, i.e., $\mathbf{E}_\pi[D_{\mathrm{TV}}(P||\hat{P})] \leq \mathcal{E}_{\hat{P}}$. Building on these assumptions and the analysis developed in Appendix A, the upper bound for the prediction error is derived as $\epsilon_k \leq \mathcal{E}_{\sigma,k}$.

We now assess the direct impact of the prediction error on the performance gap in Equation (9). As shown in Equation (2), the value function depends on both the cumulative rewards by $K$-step rollout using the pretrained dynamics model and the Q-function at the terminal state. Prediction errors in the state representation cause the policy to select sub-optimal actions, impacting both immediate rewards and future state transitions. These errors accumulate over time, distorting the terminal state and degrading the Q-function evaluation. Moreover, the pretrained policy $\pi$ was optimized for the pretraining tasks and may no longer select optimal actions in the new task due to differences in task objectives and environment dynamics. As a result, the performance gap arises from the compounded effect of both the prediction errors and the sub-optimality of the planning policy. To quantify this, we now derive an upper bound for the resulting performance drop.

Let $\Gamma := \frac{1-\gamma^{K-1}}{1-\gamma}L_r(1+L_\pi) + \gamma^K L_Q(1+L_\pi)$, $\mathcal{E}_{\max} = \max_t \mathcal{E}_{\delta,t}$ and $L_Q = L_r/(1-\gamma)$. Meanwhile, we denote the policy shift between the pretrained policy $\pi$ and the underlying optimal policy $\hat{\pi}$ for the current task to be $\mathcal{E}_\pi := \max_x D_{\mathrm{TV}}(\pi|\hat{\pi})$. Then we obtain the following result.

**Theorem 1** *Given Assumptions 1, 2 and 3 hold, the performance gap, denoted as $\eta - \hat{\eta}$, is upper bound by:*

$$\eta - \hat{\eta} \leq \left(\gamma^K \frac{\mathcal{E}_{\max}}{1-\gamma^K} + \Gamma\frac{2\gamma\mathcal{E}_{\hat{P}}}{1-\gamma^K}\right) + \Gamma\left(\frac{4r_{\max}\mathcal{E}_\pi}{1-\gamma} + \frac{4\gamma\mathcal{E}_\pi}{1-\gamma^K}\right).$$

**Determine the Dominating Mismatch.** The upper bound in Theorem 1 highlights two primary root causes for performance degradation: the mismatch of dynamics model, represented by term $\mathcal{E}_{\max}$ and $\mathcal{E}_{\hat{P}}$, where the dynamics model failing to accurately capture the true dynamics of the current task; and mismatch of policy ($\mathcal{E}_\pi$), when the pretrained policy being sub-optimal. As demonstrated in the motivating example Figure 1, effective finetuning hinges on identifying the *dominating root cause* of the performance degradation and prioritizing on its finetuning. Based on the above theoretic analysis, we have the following criterion to determine the dominating mismatch at each step:

- Update dynamics model if $\mathcal{E}_{\hat{P}} \geq C_1\mathcal{E}_\pi - C_2$, where $C_1 = \left(2r_{\max}(1-\gamma^k)/(\gamma - \gamma^2) + 2\right)$ and $C_2 = \frac{\gamma^{K-1}\mathcal{E}_{\max}}{2\Gamma}$ which implies that the errors from the dynamics model are the dominating cause of performance degradation, and improving the model's accuracy will most effectively reduce the performance gap.

- Update planning policy if $\mathcal{E}_\pi \geq \frac{1}{C_1}\mathcal{E}_{\hat{P}} + \frac{C_2}{C_1}$ which indicates that the performance degradation is more sensitive to suboptimal actions chosen by the policy, and refining the policy will be the most impactful step toward performance improvement.

Specifically, in the implementation of AdaWM (as outlined in Algorithm 1), we use the TV distances between the state distributions $P$ and $\hat{P}$ to estimate the dynamics model mismatch following Janner et al. (2019) and the policy distributions $\pi$ and $\hat{\pi}$ for the policy mismatch. In particular, at each step, if $D_{\mathrm{TV}}(P|\hat{P}) > C \cdot D_{\mathrm{TV}}(\pi|\hat{\pi})$ where $C$ is a function of $C_1$ and $C_2$, the dynamics model is updated; otherwise, the policy is updated. It is worth noting that this simplified criteria is in line with the theoretical insights while reducing computational complexity. The proof of Theorem 1 can be found in Appendix B.

## 2.2 ADAWM: MISMATCH IDENTIFICATION AND ALIGNMENT-DRIVEN FINETUNING

Based on the above analysis, we propose AdaWM with efficient finetuning strategy while minimizing performance degradation. As outlined in Algorithm 1, AdaWM consists of two key components: mismatch identification and adaptive finetuning.

---

**Algorithm 1** AdaWM: Adaptive World Model based Planning

---

**Require:** Pretrained dynamics model $\texttt{WM}_\phi(P)$ and policy $\pi_\omega$ (parameter $\Omega$). Planning horizon $K$.
 Threshold $C$. Reply buffer $\mathcal{B}$ collected from pretraining phase.
1: **for** finetuning step $t = 1, 2, \cdots$ **do**
2:      Collect samples $\mathcal{W} = \{(x, a, r)\}$ by following current policy $\pi_t$ (parameter $\omega_t$) and dynamics model $\texttt{WM}_t(P)$ (parameter $\phi_t$).
3:      Mismatch Identification: Evaluate the policy mismatch by using samples from $\mathcal{B}$ and $\mathcal{W}$ to compute the TV distance as $D_{\text{TV}}(\pi_t|\pi_\omega) \approx \max_x \|\pi_t(a|x) - \pi_\omega(a|x)\|$.
         Evaluate the mismatch of the dynamics model by $D_{\text{TV}}(P|\hat{P}) \approx \|P(x, a) - \hat{P}(x, a)\|_1$.
4:      **if** $D_{\text{TV}}(P|\hat{P}) > C \cdot D_{\text{TV}}(\pi_t|\pi_\omega)$ **then**
5:         Update dynamics model $B' \leftarrow B$, $\phi_t = (B'Z)^\top \Phi$.
6:      **else**
7:         Update policy $\Delta' \leftarrow \Delta$, $\omega_t = (\Delta')^\top \Omega$.
8:      **end if**
9: **end for**

---

**Mismatch Identification.** The first phase of AdaWM is dedicated to identifying the dominant mismatch between the pretrained task and the current task and two main types of mismatches are evaluated (line 3 in Algorithm 1): 1) Mismatch of Dynamics Model. AdaWM estimates the Total Variation (TV) distance between dynamics model by measuring state-action visitation distribution Janner et al. (2019). This metric helps quantify the model's inability to predict the current task's state accurately, revealing weaknesses in the pretrained model's understanding of the new task; and 2) Mismatch of Policy. AdaWM calculates the state visitation distribution shift using TV distance between the state visitation distributions from pretraining and the current task.

**Alignment-driven Finetuning.** Once the dominant mismatch is identified, AdaWM selectively finetunes either the dynamics model or the policy based on which component contributes more to the performance gap. In particular, AdaWM uses the following finetuning method to further reduce computational overhead and ensures efficiency (line 4-8 in Algorithm 1). 1) Finetuning the Dynamics Model. AdaWM leverages a LoRA-based Hu et al. (2021) low-rank adaptation strategy (as known as NoLa Koohpayegani et al. (2024)) to efficiently update the dynamics model. The model parameters are decomposed into two lower-dimensional vectors: the latent representation base vector $Z$ and vector $\Phi$. During finetuning, only the weight $B$ of the base vector is updated, i.e., $B' \leftarrow B$, $\phi' = (B'Z)^\top \Phi$. 2) Finetuning Planning Policy. AdaWM decomposes the policy network $\omega$ into a convex combination of sub-units $\Omega = \sum_{i=1}^{D} \delta_i \omega_i$. During finetuning, only the weight vector $\Delta = [\delta_1, \cdots, \delta_D]$ of these sub-units are updated, i.e., $\Delta' \leftarrow \Delta$.

## 3 EXPERIMENTAL RESULTS

In this section, we evaluate the effectiveness of AdaWM by addressing the following two questions: 1) Can AdaWM help to effectively mitigate the performance drop through finetuning in various CARLA tasks? 2) How does the parameter $C$ in AdaWM impact the finetuning performance.

**Experiments Environment.** We conduct our experiments in CARLA, an open-source simulator with high-fidelity 3D environment Dosovitskiy et al. (2017). At each time step $t$, agent receives bird-eye-view (BEV) as observation, which unifies the multi-modal information Liu et al. (2023); Li et al. (2023). Furthermore, by following the planned waypoints, agents navigate the environment by executing action $a_t$, such as acceleration or brake, and receive the reward $r_t$ from the environment. We define the reward as the weight sum of five attributes: safety, comfort, driving time, velocity and distance to the waypoints. The details of the CARLA environment and reward design are relegated to Appendix C.

**Baseline Algorithms.** In our study, we consider three state-of-the-art autonomous driving algorithms as baseline and evaluate their performance when deployed in the new task. Notably, we choose supervised learning based method VAD Jiang et al. (2023) and UniAD Hu et al. (2023) and use the provided checkpoints trained on Bench2Drive Jia et al. (2024) dataset for closed-loop evaluation in CARLA. Meanwhile, we also compare AdaWM with the state-of-the-art DreamerV3 based learning algorithms adopted by Think2drive Li et al. (2024).

| Algorithm | Pre-RTM03 TTC↑ | SR↑ | ROM03 TTC↑ | SR↑ | RTD12 TTC↑ | SR↑ | LTM03 TTC↑ | SR↑ | LTD03 TTC↑ | SR↑ |
|---|---|---|---|---|---|---|---|---|---|---|
| UniAD | 1.25 | 0.48 | 0.92 | 0.13 | 0.07 | 0.05 | 0.05 | 0.04 | 0.03 | 0.04 |
| VAD | 0.96 | 0.55 | 0.95 | 0.15 | 0.15 | 0.12 | 0.05 | 0.10 | 0.09 | 0.10 |
| DreamerV3 | 1.16 | 0.68 | 0.95 | 0.40 | 0.42 | 0.32 | 0.25 | 0.28 | 0.15 | 0.35 |
| **AdaWM** | 1.16 | 0.68 | **2.05** | **0.82** | **1.25** | **0.66** | **1.32** | **0.72** | **1.92** | **0.70** |

Table 2: The impact of finetuning in CARLA tasks. (RO/RT/LT: Roundabout / Right Turn / Left Turn, M/D: Moderate / Dense traffic, 03 and 12 indicate different Towns.)

**Pretrain-Finetune.** In our experiments, we use the tasks from CARLA leaderboard v2 and Bench2Drive Jia et al. (2024) for pretraining. The pretraining is conducted for 12 hour training on a single V100 GPU. After obtaining the pretrained model and policy, we conduct finetuning phase for one hour on a single V100 GPU. It is important to note that the three baseline algorithms were originally designed for offline learning, where they were trained on a fixed offline dataset and expected to generalize well to new tasks. In our comparison, we adhere to the original implementation of these baseline algorithms and evaluate their performance using the provided offline-trained checkpoints, as described in their respective papers. While finetuning is not applied to the baseline algorithms due to their offline nature, this allows us to maintain consistency with their intended design and ensure a valid comparison of their performance.

## 3.1 Performance Comparisons

In this section, we compare the learning performance among our proposed AdaWM and the baseline algorithms in terms of the time-to-collision (TTC) and success rate (SR,, also known as completion rate), i.e., the percentage of trails that the agent is able to achieve the goal without any collisions.

**Evaluation Tasks.** In our experiments, we evaluate the proposed AdaWM and the baseline methods on a series of *increasingly difficult* autonomous driving tasks. These tasks are designed to assess each model's ability to generalize to the new task and traffic condition. The first task closely mirrors the pretraining scenario, while subsequent tasks introduce more complexity and challenge. In particular, during pretraining, AdaWM is trained on Pre-RTM03 task, which involves a Right Turn in Moderate traffic in Town 03. Following the pretraining, we evaluate the learning performance in four tasks, respectively: 1) Task ROM03: This task is a ROundabout in Moderate traffic in Town 03. While it takes place in the same town as the pretraining task, the introduction of a roundabout adds complexity to the driving scenario; 2) Task RTD12: This task features a Right Turn in Dense traffic in Town 12. The increased traffic density and different town environment make this task more challenging than the pretraining task; 3) Task LTM03: This task involves a Left Turn in Moderate traffic in Town 03. Although it takes place in the same town and traffic conditions as the pretraining task, the switch to a left turn introduces a new challenge; and 4) Task LTD03: The most challenging task as it involves a Left Turn in Dense traffic in Town 03. The combination of heavy traffic and a left turn in a familiar town environment makes this task the hardest in the evaluation set. We summarize the evaluation results in Table 2 and Table 3. We also include the learning curve in Figure 3. Meanwhile, we also include another five challenging scenarios for verify the capability of AdaWM in Appendix G.

### 3.1.1 Evaluation Results

**The Impact of Finetuning in Autonomous Driving.** Following previous works on the effectiveness of fine-tuning in robotic learning Julian et al. (2021); Feng et al. (2023), we first validate this finding in the autonomous driving domain. As shown in Table 2, AdaWM consistently outperforms its pretrained only counterparts and two supervised learning based methods, i.e., VAD and UniAD, across various tasks. Starting with Task ROM03, which closely resembles the pretraining scenario Pre-RTM03 (right turn, moderate traffic in Town 03), our method achieved a TTC of 2.05 and an SR of 0.82. This far exceeds the performance of the baseline methods, where DreamerV3, VAD, and UniAD achieved TTC values of 0.95, 0.95, and 0.92, respectively, and much lower success rates, with the highest being 0.40 by DreamerV3. This significant difference highlights AdaWM's ability to adapt more effectively even in familiar environments. In the most challenging task, LTD03 (left turn with dense traffic in Town 03), AdaWM continues to excel, achieving a TTC of 1.92 and an SR

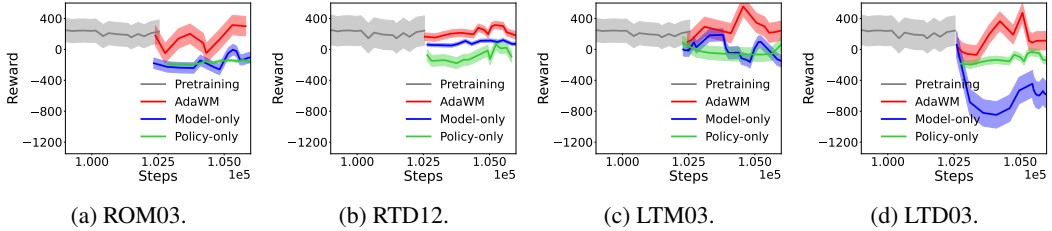

Figure 3: Learning curves of different finetuning strategies in four evaluation tasks.

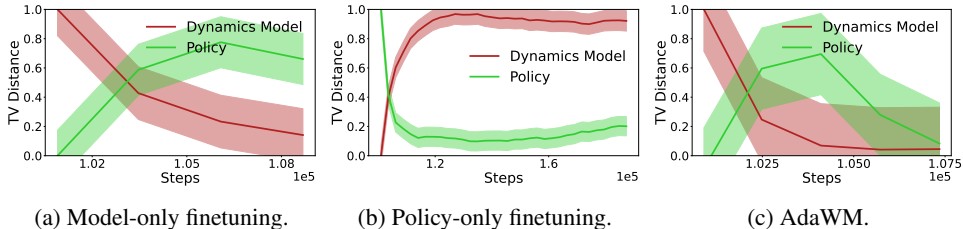

Figure 4: The mismatches of the dynamics model and policy during the finetuning.

of 0.70. In contrast, DreamerV3, the best-performing baseline, reached only 0.15 in TTC and 0.35 in SR, while both VAD and UniAD struggled with TTC values below 0.1 and SRs as low as 0.04. Similar trends are observed in Tasks RTD12 (right turn, dense traffic in Town 12) and LTM03 (left turn, moderate traffic in Town 03), where AdaWM consistently outperforms the baseline methods, achieving both higher TTC and SR scores. These results affirm that AdaWM's adaptive finetuning approach significantly improves performance across various challenging tasks, ensuring both safer and more reliable decision-making.

**Effectiveness of Finetuning Strategy.** Building upon the established benefits of finetuning, we next demonstrate the superiority of our alignment-driven finetuning strategy compared to standard finetuning methods in Table 3. While conventional approaches like model-only, policy-only and alternate finetuning show improvements over pretraining, AdaWM's alignment-driven strategy consistently achieves better performance. For instance, in Task ROM03 (roundabout, moderate traffic in Town03), AdaWM achieved a TTC of 2.05 and an SR of 0.82, surpassing the model-only (TTC 0.95, SR 0.60) and policy-only (TTC 0.46, SR 0.72) approaches. These results demonstrate that AdaWM's finetuning strategy, which adaptively adjusts the model or policy based on the results from mismatch identification, provides the most robust performance in complex driving scenarios.

## 3.2 ABLATION STUDIES

**Determine the Dominating Mismatch during Finetuning.** We compare the changes of two mismatches, mismatch of dynamics model and mismatch of policy, for different finetuning strategies in Figure 4. It can be seen from Figure 4a that model-only finetuning often leads to a deterioration in policy performance, as indicated by a significant increase in TV distance. This suggests that while the model adapts to new task, the policy struggles to keep pace, resulting in suboptimal decision-making. Conversely, as shown in Figure 4b, policy-only finetuning reduces the TV distance between policies, but this comes at the cost of an increased mismatch of dynamics model, signaling a growing discrepancy between the learned dynamics model and the actual environment. In contrast, in Figure 4c, we show that our proposed alignment-driven finetuning method in AdaWM can effectively align both factors in the new task. By selectively adjusting the model or policy at each step, this adaptive method prevents either error from escalating dramatically, maintaining stability and ensuring better performance throughout the finetuning process.

**The Impact of Parameter $C$.** In Table 4, we study the effect of different parameter $C$ on four tasks (ROM03, RTD12, LTM03, LTD03) in terms of TTC and SR. Notably, when $C$ becomes too large, AdaWM's performance deteriorates, as it essentially reduces to policy-only finetuning. This is reflected in the sharp drop in both TTC and SR for high $C$ values, such as $C = 100$, across all tasks.

| Algorithm | ROM03 TTC ↑ | SR↑ | RTD12 TTC ↑ | SR↑ | LTM03 TTC ↑ | SR ↑ | LTD03 TTC↑ | SR↑ |
|---|---|---|---|---|---|---|---|---|
| No finetuning | 0.95 | 0.40 | 0.42 | 0.32 | 0.25 | 0.28 | 0.15 | 0.35 |
| Policy-only | 0.46 | 0.72 | 1.21 | 0.63 | 0.21 | 0.61 | 0.62 | 0.61 |
| Model-only | 0.95 | 0.60 | 0.83 | 0.48 | **1.39** | 0.68 | 1.49 | 0.63 |
| Model+Policy | 0.72 | 0.52 | 0.92 | 0.50 | 1.09 | 0.60 | 1.21 | 0.58 |
| **AdaWM** | **2.05** | **0.82** | **1.25** | **0.66** | 1.32 | **0.72** | **1.92** | **0.70** |

Table 3: Comparison on the effectiveness of different finetuning strategies.

| $C$ | ROM03 TTC↑ | SR↑ | RTD12 TTC↑ | SR↑ | LTM03 TTC ↑ | SR↑ | LTD03 TTC↑ | SR↑ |
|---|---|---|---|---|---|---|---|---|
| 0.5 | 1.23 | 0.58 | 1.16 | 0.50 | 1.15 | 0.47 | 1.27 | 0.52 |
| 2 | 1.82 | 0.70 | 1.20 | 0.57 | 1.28 | 0.68 | 1.73 | 0.63 |
| 5 | 2.05 | 0.82 | **1.25** | **0.66** | **1.32** | **0.72** | 1.92 | 0.70 |
| 10 | **2.15** | **0.85** | **1.25** | 0.62 | 1.24 | 0.62 | **2.01** | **0.72** |
| 50 | 1.86 | 0.71 | 0.87 | 0.51 | 1.30 | 0.68 | 1.78 | 0.62 |
| 100 | 1.17 | 0.45 | 0.62 | 0.46 | 0.92 | 0.43 | 1.05 | 0.32 |

Table 4: Ablation studies on $C$.

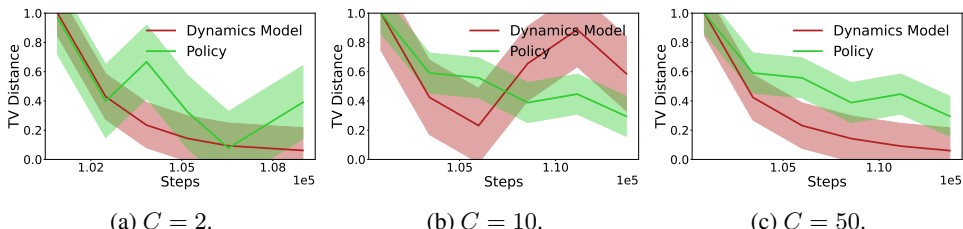

(a) $C = 2$.  (b) $C = 10$.  (c) $C = 50$.

Figure 5: The mismatches of the dynamics model and policy with different value of $C$.

On the other hand, very small $C$ values result in suboptimal performance due to insufficient updates to the dynamics model, underscoring the importance of alignment-driven finetuning for achieving robust learning. Meanwhile, the results demonstrate that AdaWM performs well across a wide range of $C$ values (between 2 and 50). As further illustrated in Figure 5, AdaWM effectively controls mismatches in both the policy and the dynamics model for $C = 2, 10$, and $50$. For instance, in Figure 5b, when the mismatch in the dynamics model becomes more pronounced than in the policy, AdaWM prioritizes fine-tuning the dynamics model, which in turn helps reduce policy sub-optimality and preserves overall performance. Conversely, in Figure 5a and Figure 5c, when the policy mismatch is more dominant, AdaWM identifies and fine-tunes the policy accordingly. These results emphasize AdaWM's adaptability in managing different sources of mismatch, ensuring efficient fine-tuning and strong performance across a variety of tasks.

## 4 CONCLUSION

In this work, we propose AdaWM, an Adaptive World Model based planning method that mitigates performance drops in world model based reinforcement learning (RL) for autonomous driving. Building on our theoretical analysis, we identify two primary causes for the performance degradation: mismatch of the dynamics model and mismatch of the policy. Building upon our theoretical analysis, we propose AdaWM with two core components: mismatch identification and alignment-driven finetuning. AdaWM evaluates the dominating source of performance degradation and applies selective low-rank updates to the dynamics model or policy, depending on the identified mismatch. Extensive experiments on CARLA demonstrate that AdaWM significantly improves both route success rate and time-to-collision, validating its effectiveness. This work emphasizes the importance of choosing an efficient and robust finetuning strategy in solving challenging real-world tasks. There are several promsing avenues for future research. First, exploring the generalization of AdaWM to other domains beyond autonomous driving could broaden its applicability. Additionally, extending AdaWM to multi-agent setting that accounts for the interaction among agents could further enhance its robustness in complex real-world environments.

ACKNOWLEDGMENT

We acknowledge that JSZ is generously supported in part by NSF Grants CNS-2203239, ECCS-2413529, CCSS-2203238, and ARO Grant W911NF-2410046. We also would like to express our great appreciation to all reviewers for their constructive comments and feedback to help us improve our work.

ETHICS STATEMENT

AdaWM is developed to improve finetuning in world model based reinforcement learning, specifically for autonomous driving applications. The focus is on addressing performance degradation due to distribution shifts, ensuring safer and more reliable decision-making in dynamic environments. While AdaWM aims to enhance the adaptability and robustness of autonomous systems, ethical considerations are paramount. These include ensuring that the system operates safely under real-world conditions, minimizing unintended biases from pretraining data, and maintaining transparency in how decisions are made during finetuning. Additionally, the societal implications of deploying autonomous driving technologies, such as their impact on public safety and employment, require ongoing attention. Our commitment is to ensure that AdaWM contributes positively and responsibly to the future of autonomous driving systems.

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

# Appendix

## A  PROOF OF UPPER BOUND OF PREDICTION ERROR

We first present the results on the upper bound of the prediction error in Lemma 1 below. For brevity, we denote $M = B_V B_U \frac{(B_W)^k - 1}{B_W - 1}$, $\Psi_k(\delta, n) = l_n + 3\sqrt{\frac{\log\left(\frac{2}{\delta}\right)}{2n}} + \mathcal{O}\left(d\frac{MBa(1+\sqrt{2\log(2)k})}{\sqrt{n}}\right)$, where $d$ is the dimension of the latent state representation and $N_1 = L_h L_z L_a UV$, $N_2 = L_h L_z VW + L_z VA$. Then we obtain the following result on the upper bound of the prediction error.

**Lemma 1** *Under Assumptions 1 and 2, we have that with probability at least $1 - \delta$, the prediction error $\epsilon_k$, for $k \geq 1$, is upper bounded by*

$$\epsilon_K \leq \sum_{j=1}^{K} N_1^j \left( \sqrt{\Psi_h(\delta, n)} + 1/\delta(N_2 \mathcal{E}_x + 2h B_x(\mathcal{E}_{\hat{P}} - \mathcal{E}_P)) \right) := \mathcal{E}_{\delta, K}$$

*Proof.* We decompose the prediction error into two terms,

$$\epsilon_k = (x_k - \bar{x}_k) + (\bar{x}_k - \hat{x}_k) \tag{5}$$

where $\bar{x}_k$ is the underlying true state representation.

- $(x_k - \bar{x}_k)$: distribution shift between the pretraining task and the new task
- $(\bar{x}_k - \hat{x}_k)$: the generalization error of the pre-trained RNN model on the pre-training task.

Next, we derive the upper bound for each term respectively.

**Upper bound for $(x_k - \bar{x}_k)$.** Let the expected total-variation distance between the pretraining task to be $\hat{P}(x'|x, a)$ and the new task to be $P(x'|x, a)$ be upper bounded by $\mathcal{E}_P$, i.e., $\mathbf{E}_\pi[D_{\mathrm{TV}}(P||\bar{P})] \leq \mathcal{E}_P$.

Following the same line as in Lemma B.2 Janner et al. (2019), we assume

$$\max_t E_{x \sim P^t(z)} D_{KL} \left( P\left(x' \mid x\right) \| \bar{P}\left(x' \mid x\right) \right) \leq \mathcal{E}_P,$$

and the initial distributions are the same.

Then we have the marginal state visitation probability is upper bounded by

$$\frac{1}{2} \sum_x |\rho^{t+k}(x) - \bar{\rho}^{t+k}(x)| \leq k\mathcal{E}_P.$$

Meanwhile, for simplicity, we define the following notations to characterize the prediction error at $k$ time step.

$$\begin{aligned}
\mathbf{E}[x^{t+k}] &= \rho^k \geq 0, \\
\mathbf{E}[\bar{x}^{t+k}] &= \bar{\rho}^k \geq 0, \\
\bar{\epsilon}_k &:= x^k - \bar{x}^k,
\end{aligned}$$

where $\rho^k = \mathbf{E}_{x \sim \rho^k(x)}[x] = \sum_x x\rho^k(x)$ is the mean value of the marginal visitation distribution at time step $k$ (starting from time step 0).

Then we obtain the upper bound for the non-stationary part of the prediction error as follows,

$$\mathbf{E}[\bar{\epsilon}_k] := \rho^k - \bar{\rho}^k \leq 2k B_x \mathcal{E}_P$$

**Upper bound for $(\bar{x}_k - \hat{x}_k)$.** We consider the setting where RNN model is obtained by training on $n$ i.i.d. samples of state-action-state sequence $\{x_t, a_t, x_{t+1}\}$ and the empirical loss is $l_n$ with loss function $f$. Denote $\epsilon_k^{\mathrm{RNN}} := \bar{x}_k - \hat{x}_k$. The RNN is trained to map the one step input, i.e., $x_t, a_t$, to the output $x_{t+1}$. Particularly, the world model leverage the RNN to make prediction over the future steps. Following the standard probably approximately correct (PAC) learning analysis framework, we first recall the following results Wu et al. (2021) on the RNN generalization error. For brevity, we denote $M = B_V B_U \frac{(B_W)^k - 1}{B_W - 1}$, $\Psi_k(\delta, n) = l_s + 3\sqrt{\frac{\log\left(\frac{2}{\delta}\right)}{2n}} + \mathcal{O}\left(d\frac{MBa(1+\sqrt{2\log(2)k})}{\sqrt{n}}\right)$, where $d$ is the dimension of the latent state representation

**Lemma 2 (Generalization Error of RNN)** *Assume the weight matrices satisfy Assumption 1 and the input satisfies Assumption 2. Assume the training and testing datasets are drawn from the same distribution. Then with probability at least $1 - \sigma$, the generalization error in terms of the expected loss function has the upper bound as follows,*

$$\mathbf{E}[f(x_{i,t+k} - \bar{x}_{i,t+k}] \leq l_n + 3\sqrt{\frac{\log\left(\frac{2}{\delta}\right)}{2n}} + \mathcal{O}\left(L_r d_y \frac{dMB_a(1 + \sqrt{2\log(2)k})}{\sqrt{n}}\right)$$

In particular, the results in Lemma 2 considers the least square loss function and the generalization bound only applies to the case when the data distribution remains the same during the testing. In our case, since the testing and training sets are collected from the same simulation platform thus following the same dynamics. For simplicity, in our problem setting, we assume the underlying distribution of input $\{x_t, a_t\}$ is assumed to be uniform. Subsequently, we establish the upper bound for the generalization error. Let $\epsilon_{t+k} = x_{i,t+k} - \bar{x}_{i,t+k}$, then we have, with probability at least $1 - \delta$,

$$\epsilon_k^{\text{RNN}} \leq \sqrt{\Psi_k(\delta, n)} \tag{6}$$

**Error Accumulation and Propagation.** We first recall the decomposition of the prediction error as follows.

$$\epsilon_k = \epsilon_k^{\text{RNN}} + \bar{\epsilon}_k \tag{7}$$

Then by using the Lipschitz properties of the activation functions and Assumption 2, we obtain the upper bound for the error,

$$\begin{aligned}
\epsilon_k =& \epsilon_k^{\text{RNN}} + \bar{\epsilon}_k \\
\leq& \epsilon_{t+k}^{\text{RNN}} + L_h L_z V(W\epsilon_{x,t+k} + UL_a\epsilon_{x,t+k-1}) \\
=& \epsilon_{t+k}^{\text{RNN}} + L_h L_z VW\epsilon_{x,t+k} + L_h L_z VUL_a\epsilon_{x,t+k-1} \\
=& \epsilon_{t+k}^{\text{RNN}} + (L_h L_z VW + L_z VA)\epsilon_{i,x,t+k} + L_h L_z VUL_a\epsilon_{i,x,t+k-1} \\
:=& M_{t+k} + N\epsilon_{x,t+k-1}
\end{aligned} \tag{8}$$

where

$$\begin{aligned}
M_{t+k} :=& \epsilon_{t+k}^{\text{RNN}} + (L_h L_z VW + L_z VA)\epsilon_{x,t+k} \\
N :=& L_h L_z VUL_a.
\end{aligned}$$

Notice that in the stationary part of the prediction error, we have $|\epsilon_{x,t+k}| = |\epsilon_{t+k}|$. Furthermore, by abuse of notation, we apply Equation (8) recursively and obtain the relationship between the prediction error at $k$ rollout horizon and the gap from the input, i.e.,

$$\begin{aligned}
\epsilon_{t+k} \leq& M_{t+k} + N\epsilon_{t+k-1} \\
\leq& M_{t+k} + NM_{t+k-2} \\
\leq& \cdots \\
\leq& \sum_{h=0}^{k} N^h M_{t+k-h} + N^k \epsilon_t
\end{aligned}$$

Taking expectation on both sides gives us,

$$\begin{aligned}
\mathbf{E}[\epsilon_{t+k}] \leq& \sum_{h=0}^{k} N^h \mathbf{E}[M_{t+k-h}] + N^k \mathbf{E}[\epsilon_t] \\
\leq& \sum_{h=0}^{k} N^h (2hB_x \mathcal{E}_P + N_1 \mathcal{E}_x + \mathbf{E}[\epsilon_{t+h}^{\text{RNN}}]),
\end{aligned}$$

where $N_1 = L_h L_z VW + L_z VA$.

**The Upper Bound of the Prediction Error.** Then by invoking Markov inequality, we have the upper bound for $\epsilon_t$ with probability at least $1 - \delta$ as follows,

$$\epsilon_{t+k} \leq \sum_{h=1}^{k} N^h \left( \frac{1}{\delta}(2hB_x\mathcal{E}_P + N_1\mathcal{E}_x) + \sqrt{\Psi_t(n, \delta)} \right) := \mathcal{E}_{\delta, t}$$

# B    PROOF OF THEOREM 1

Next, we quantify the performance gap when using the pre-trained policy and model in the current task.

$$\eta - \hat{\eta} = \mathbf{E}_{z\sim P, a\sim\pi_\omega, \mathtt{WM}_\phi} \left[ \sum_{i=1}^{K} \gamma^i r(z_i, a_i) + Q(z_K, a_K) \right] - \mathbf{E}_{z\sim\hat{P}, a\sim\pi_\omega, \mathtt{WM}_\phi} \left[ \sum_{i=1}^{K} \gamma^i r(z_i, a_i) + Q(z_K, a_K) \right]$$

$$= \left( \mathbf{E}_{z\sim P, a\sim\pi_\omega, \mathtt{WM}_\phi} \left[ \sum_{i=1}^{K} \gamma^i r(z_i, a_i) \right] - \mathbf{E}_{z\sim\hat{P}, a\sim\pi_\omega, \mathtt{WM}_\phi} \left[ \sum_{i=1}^{K} \gamma^i r(z_i, a_i) \right] \right)$$

$$+ \gamma^K \left( \mathbf{E}_{z\sim P, a\sim\pi_\omega, \mathtt{WM}_\phi} [Q(z_K, a_K)] - \mathbf{E}_{z\sim\hat{P}, a\sim\pi_\omega, \mathtt{WM}_\phi} [Q(z_K, a_K)] \right) \quad (9)$$

Note the first term on the RHS is associated with the modeling error and the sub-optimality of the policy, while the second term is only relevant to modeling error. In what follows, we recap the formulation for world-model based RL.

Then, by follow the same line as in Janner et al. (2019), let $\Gamma := \frac{1-\gamma^{L-1}}{1-\gamma} L_r(1+L_\pi) + \gamma^L L_Q(1+L_\pi)$, $\mathcal{E}_{\max} = \max_t \mathcal{E}_{\delta, t}$ and $L_Q = L_r/(1-\gamma)$. Meanwhile, we denote the policy divergence to be $\mathcal{E}_\pi := \max_z D_{\mathrm{TV}}(\pi|\hat{\pi})$. Let the expected total-variation distance between the true state transition probability $\hat{P}(z'|z, a)$ and the predicted (current tasks) one $P(z'|z, a)$ be upper bounded by $\mathcal{E}_{\hat{P}}$, i.e., $\mathbf{E}_\pi[D_{\mathrm{TV}}(\tilde{P}||\hat{P})] \leq \mathcal{E}_{\hat{P}}$. Then we obtain the first term is upper bounded by

**Lemma 3** *Given Assumption 3 holds, the first term in Equation* (9) *is upper bounded by,*

$$\Gamma \left( \frac{2\gamma(\mathcal{E}_{\hat{P}} + 2\mathcal{E}_\pi)}{1 - \gamma^K} + \frac{4r_{\max}\mathcal{E}_\pi}{1 - \gamma} \right) \quad (10)$$

By using the definition of Q-function and the upper bound of the prediction error in Lemma 1, we obtain that the second term in Eqn. 9 is upper bounded by

$$\gamma^K \frac{\mathcal{E}_{\max}}{1 - \gamma}$$

By combining the upper bounds of the two terms, we obtain the upper bound in Theorem 1.

## B.1    DERIVATION OF $C_1$ AND $C_2$

Next, we derive the parameter $C_1$ and $C_2$ in AdaWM (ref. Section 2.1).

The RHS of the inequality in Theorem 1 can be divided into two parts, i.e.,

$$\left( \gamma^K \frac{\mathcal{E}_{\max}}{1 - \gamma^K} + \Gamma \frac{2\gamma\mathcal{E}_{\hat{P}}}{1 - \gamma^K} \right) + \Gamma \left( \frac{4r_{\max}\mathcal{E}_\pi}{1 - \gamma} + \frac{4\gamma\mathcal{E}_\pi}{1 - \gamma^K} \right),$$

where the first part is relevant to the dynamics model mismatch and the second part is related to the policy mismatch. Evidently, when the first term is larger than the second one, we have the dynamics mismatch to be more dominant, i.e.,

$$\left( \gamma^K \frac{\mathcal{E}_{\max}}{1 - \gamma^K} + \Gamma \frac{2\gamma\mathcal{E}_{\hat{P}}}{1 - \gamma^K} \right) > \Gamma \left( \frac{4r_{\max}\mathcal{E}_\pi}{1 - \gamma} + \frac{4\gamma\mathcal{E}_\pi}{1 - \gamma^K} \right)$$

$$\rightarrow \mathcal{E}_{\hat{P}} \geq \left( 2r_{\max}(1 - \gamma^k)/(\gamma - \gamma^2) + 2 \right) \mathcal{E}_\pi - \frac{\gamma^{K-1}\mathcal{E}_{\max}}{2\Gamma} := C_1\mathcal{E}_\pi - C_2,$$

where $C_1 = \left(2r_{\max}(1-\gamma^k)/(\gamma-\gamma^2)+2\right)$ and $C_2 = \frac{\gamma^{K-1}\mathcal{E}_{\max}}{2\Gamma}$.

Similarly, we obtain that when $\mathcal{E}_\pi \geq \frac{1}{C_1}\mathcal{E}_{\hat{P}} + \frac{C_2}{C_1}$, the policy mismatch is more dominant. To summarize, we have,

- Update dynamics model if $\mathcal{E}_{\hat{P}} \geq C_1\mathcal{E}_\pi - C_2$, where $C_1 = \left(2r_{\max}(1-\gamma^k)/(\gamma-\gamma^2)+2\right)$ and $C_2 = \frac{\gamma^{K-1}\mathcal{E}_{\max}}{2\Gamma}$ which implies that the errors from the dynamics model are the dominating cause of performance degradation, and improving the model's accuracy will most effectively reduce the performance gap.

- Update planning policy if $\mathcal{E}_\pi \geq \frac{1}{C_1}\mathcal{E}_{\hat{P}} + \frac{C_2}{C_1}$ which indicates that the performance degradation is more sensitive to suboptimal actions chosen by the policy, and refining the policy will be the most impactful step toward performance improvement.

## C  EXPERIMENTS DETAILS

**CARLA Environment**  The vehicle's state consists of two primary sources of information: environmental observations and the behavior of surrounding vehicles. Environmental observations are captured through sensors such as cameras, radar, and LiDAR, providing information about objects in the environment and their geographical context. Following standard approaches Bansal et al. (2018); Chen et al. (2023), we represent the vehicle's state using a bird's-eye view (BEV) semantic segmentation image of size $128 \times 128$.

In our experiments, we use a discrete action space. At each time step, the agent selects both acceleration and steering angle. The available choices for acceleration are $[-2, 0, 2]$, and for steering angle are $[-0.6, -0.2, 0, 0.2, 0.6]$.

We design the reward as the weighted sum of six different factors, i.e.,

$$R_t = w_1 R_{\text{safe}} + w_2 R_{\text{comfort}} + w_3 R_{\text{time}} + w_4 R_{\text{velocity}} + w_5 R_{\text{ori}} + w_6 R_{\text{target}},$$

In particular,

- $R_{\text{safe}}$ is the time to collision to ensure safety
- $R_{\text{comfort}}$ is relevant to jerk behavior and acceleration
- $R_{\text{time}}$ is to punish the time spent before arriving at the destination
- $R_{\text{velocity}}$ is to penalize speeding when the velocity is beyond 5m/s and the leading vehicle is too close
- $R_{\text{ori}}$ is to penalize the large orientation of the vehicle
- $R_{\text{target}}$ is to encourage the vehicle to follow the planned waypoints

**CARLA Benchmark tasks.**  We use the same configuration for scriveners and routes as defined in the Bench2Drive dataset Jia et al. (2024). In particular, we consider the following tasks:

- SignalizedJunctionRightTurn_Town03_Route26775_Weather2 (Pre-RTM03)
- NoScenario_Town03_Route27530_Weather25 (ROM03)
- NonSignalizedJunctionRightTurn_Town12_Route7979_Weather0 (RTD12)
- SignalizedJunctionLeftTurn_Town03_Route26700_Weather22 (LTM03)
- NonSignalizedJunctionLeftTurn_Town03_Route27000_Weather23 (LTD03)

The evaluation metric is evaluated when the agent is navigating along the pre-determined waypoints. Below are the list of the tasks configuration considered in our experiments.

Pre-RTM03

```
Pre-RTM03: &SignalizedJunctionRightTurn_Town03_Route26775_Weather2
   env:
      world:
```

```
      town: Town03
      Weather: 2
      Route: 26775
    name: Pre-RTM03
    observation.enabled: [camera, collision, birdeye_wpt]
    <<: *carla_wpt
    lane_start_point: [6.0, -101.0, 0.1, -90.0]
    ego_path: [[6.0, -101.0, 0.1], [-126, 214, 0.1]]
    use_road_waypoints: [True, False]
    use_signal: True
```

### ROM03

```
ROM03: &NoScenario_Town03_Route27530_Weather25
  env:
    world:
      town: Town03
      Weather: 25
      Route: 26530
    name: ROM03
    observation.enabled: [camera, collision, birdeye_wpt]
    <<: *carla_wpt
    lane_start_point: [12.0, 21.0, 0.1, -90.0]
    ego_path: [[12.0, 21.0, 0.1], [226, 132, 0.1]]
    use_road_waypoints: [True, False]
    use_signal: True
```

### RTD12

```
RTD12: &NonSignalizedJunctionRightTurn_Town12_Route7979_Weather0
  env:
    world:
      town: Town12
      Weather: 0
      Route: 7979
    name: RTD12
    observation.enabled: [camera, collision, birdeye_wpt]
    <<: *carla_wpt
    lane_start_point: [27.0, 101.0, 0.1, -90.0]
    ego_path: [[27.0, 101.0, 0.1], [-89, 231, 0.1]]
    use_road_waypoints: [True, False]
    use_signal: False
```

### LTM03

```
LTM03: &SignalizedJunctionLeftTurn_Town03_Route26700_Weather22
  env:
    world:
      town: Town03
      Weather: 22
      Route: 26700
    name: LTM03
    observation.enabled: [camera, collision, birdeye_wpt]
    <<: *carla_wpt
    lane_start_point: [11.0, -21.0, 0.1, -90.0]
    ego_path: [[11.0, -21.0, 0.1], [-71, 127, 0.1]]
    use_road_waypoints: [True, False]
    use_signal: True
```

### LTD03

| Dimension | L |
|---|---|
| GRU recurrent units | 2048 |
| CNN multiplier | 64 |
| Dense hidden units | 768 |
| MLP layers | 4 |
| Parameters | 77M |

Table 5: Model Sizes Hafner et al. (2023).

```
LTD03:   &NonSignalizedJunctionLeftTurn_Town03_Route27000_Weather23
  env:
    world:
      town: Town03
      Weather: 23
      Route: 27000
    name: LTD03
    observation.enabled: [camera, collision, birdeye_wpt]
    <<: *carla_wpt
    lane_start_point: [9.0, -47.0, 0.1, -90.0]
    ego_path: [[9.0, -47.0, 0.1], [83, 229, 0.1]]
    use_road_waypoints: [True, False]
    use_signal: False
```

## D    TERMINOLOGY

In this work, we distinguish world model and dynamics model. The key difference lies in their scope and functionality. A *world model* Ha & Schmidhuber (2018) is a comprehensive internal representation that an agent builds to understand its environment, including not just the state transitions (dynamics) but also observations, rewards, and potentially agent intentions. It enables agents to simulate future trajectories, plan, and predict outcomes before acting. In contrast, a dynamics model is a more specific component focused solely on predicting how the environment's state will evolve based on the current state and the agent's actions. While the *dynamics model* predicts state transitions, the world model goes further by incorporating how the agent perceives the environment (observation model) and the rewards it expects to receive (reward model).

## E    WORLD MODEL TRAINING

We use Dreamer v3 Hafner et al. (2023) structure, i.e., encoder-decoder, RSSM Hafner et al. (2019), to train the world model and adopt the Large model for all experiments with dimension summarized in Table 5. We first restate the hyper-parameters in Table 6.

**Learning BEV Representation.** The BEV representation can be learnt by using algorithms such as BevFusion Liu et al. (2023), which is capable of unifying the cameras, LiDAR, Radar data into a BEV representation space. In our experiment, we leverage the privileged information provided by CARLA Dosovitskiy et al. (2017), such as location information and map topology to construct the BEVs.

| Name | Symbol | Value |
|---|---|---|
| **General** | | |
| Replay capacity (FIFO) | — | $10^6$ |
| Batch size | $B$ | 16 |
| Batch length | $T$ | 64 |
| Activation | — | $\mathrm{LayerNorm} + \mathrm{SiLU}$ |
| **World Model** | | |
| Number of latents | — | 32 |
| Classes per latent | — | 32 |
| Reconstruction loss scale | $\beta_{\mathrm{pred}}$ | 1.0 |
| Dynamics loss scale | $\beta_{\mathrm{dyn}}$ | 0.5 |
| Representation loss scale | $\beta_{\mathrm{rep}}$ | 0.1 |
| Learning rate | — | $10^{-4}$ |
| Adam epsilon | $\epsilon$ | $10^{-8}$ |
| Gradient clipping | — | 1000 |
| **Actor Critic** | | |
| Imagination horizon | $H$ | 15 |
| Discount horizon | $1/(1-\gamma)$ | 333 |
| Return lambda | $\lambda$ | 0.95 |
| Critic EMA decay | — | 0.98 |
| Critic EMA regularizer | — | 1 |
| Return normalization scale | $S$ | $\mathrm{Per}(R, 95) - \mathrm{Per}(R, 5)$ |
| Return normalization limit | $L$ | 1 |
| Return normalization decay | — | 0.99 |
| Actor entropy scale | $\eta$ | $\mathbf{3 \cdot 10^{-4}}$ |
| Learning rate | — | $3 \cdot 10^{-5}$ |
| Adam epsilon | $\epsilon$ | $10^{-5}$ |
| Gradient clipping | — | 100 |

Table 6: Dreamer v3 hyper parameters Hafner et al. (2023).

**World Model Training.** The world model is implemented as a Recurrent State-Space Model (RSSM) Hafner et al. (2019; 2023) to learn the environment dynamics, encoder, reward, continuity and encoder-decoder. We list the equations from the RSSM mode as follows:

$$
\mathrm{RSSM} \begin{cases}
\text{Sequence model:} & h_t = f_\phi(h_{t-1}, z_{t-1}, a_{t-1}) \\
\text{Encoder:} & z_t \sim q_\phi(z_t|h_t, x_t) \\
\text{Dynamics predictor:} & \hat{z}_t \sim p_\phi(\hat{z}_t|h_t)
\end{cases}
$$

$$
\begin{aligned}
\text{Reward predictor:} & \quad \hat{r}_t \sim p_\phi(\hat{r}_t|h_t, z_t) \\
\text{Continue predictor:} & \quad \hat{c}_t \sim p_\phi(\hat{c}_t|h_t, z_t) \\
\text{Decoder:} & \quad \hat{x}_t \sim p_\phi(\hat{x}_t|h_t, z_t)
\end{aligned} \tag{11}
$$

We follow the same line as in Dreamer v3 Hafner et al. (2023) to train the parameter $\phi$. We include the following verbatim copy of the loss function considered in their work.

Given a sequence batch of inputs $x_{1:T}$, actions $a_{1:T}$, rewards $r_{1:T}$, and continuation flags $c_{1:T}$, the world model parameters $\phi$ are optimized end-to-end to minimize the prediction loss $\mathcal{L}_{\mathrm{pred}}$, the dynamics loss $\mathcal{L}_{\mathrm{dyn}}$, and the representation loss $\mathcal{L}_{\mathrm{rep}}$ with corresponding loss weights $\beta_{\mathrm{pred}} = 1$, $\beta_{\mathrm{dyn}} = 0.5$, $\beta_{\mathrm{rep}} = 0.1$:

$$
\mathcal{L}(\phi) \doteq \mathbf{E}_{q_\phi} \left[ \sum_{t=1}^{T} (\beta_{\mathrm{pred}} \mathcal{L}_{\mathrm{pred}}(\phi) + \beta_{\mathrm{dyn}} \mathcal{L}_{\mathrm{dyn}}(\phi) + \beta_{\mathrm{rep}} \mathcal{L}_{\mathrm{rep}}(\phi)) \right]. \tag{12}
$$

$$
\begin{aligned}
\mathcal{L}_{\mathrm{pred}}(\phi) &\doteq -\ln p_\phi(x_t|z_t, h_t) - \ln p_\phi(r_t|z_t, h_t) - \ln p_\phi(c_t|z_t, h_t) \\
\mathcal{L}_{\mathrm{dyn}}(\phi) &\doteq \max\big(1, \mathrm{KL}[\mathrm{sg}(q_\phi(z_t|h_t, x_t)) \| \quad p_\phi(z_t|h_t) ]\big) \\
\mathcal{L}_{\mathrm{rep}}(\phi) &\doteq \max\big(1, \mathrm{KL}[ \quad q_\phi(z_t|h_t, x_t) \| \mathrm{sg}(p_\phi(z_t|h_t))]\big)
\end{aligned} \tag{13}
$$

**Actor-Critic Learning.** We consider the prediction horizon to be 16 as the same as in Dreamer v3 while training the actor-critic networks. We follow the same line as in Dreamer v3 and consider the actor and critic defined as follows.

$$
\begin{aligned}
\text{Actor:} \quad & a_t \sim \pi_\theta(a_t|x_t) \\
\text{Critic:} \quad & v_\psi(x_t) \approx \mathbf{E}_{p_\phi, \pi_\theta}[R_t],
\end{aligned} \tag{14}
$$

where $R_t \doteq \sum_{\tau=0}^{\infty} \gamma^\tau r_{t+\tau}$ with discounting factor $\gamma = 0.997$. Meanwhile, to estimate returns that consider rewards beyond the prediction horizon, we compute bootstrapped $\lambda$-returns that integrate the predicted rewards and values:

$$
R_t^\lambda \doteq r_t + \gamma c_t \Big( (1-\lambda) v_\psi(s_{t+1}) + \lambda R_{t+1}^\lambda \Big) \qquad R_T^\lambda \doteq v_\psi(s_T) \tag{15}
$$

### E.1 TRAINING DATASET: BENCH2DRIVE

In our experiments, we use the open source Bench2Drive dataset Jia et al. (2024); Li et al. (2024), which is a comprehensive benchmark designed to evaluate end-to-end autonomous driving (E2E-AD) systems in a closed-loop manner. Unlike existing benchmarks that rely on open-loop evaluations or fixed routes, Bench2Drive offers a more diverse and challenging testing environment. The dataset consists of 2 million fully annotated frames, collected from 10,000 short clips distributed across 44 interactive scenarios, 23 weather conditions, and 12 towns in CARLA. This diversity allows for a more thorough assessment of autonomous driving capabilities, particularly in corner cases and complex interactive situations that are often underrepresented in other datasets.

A key feature of Bench2Drive is its focus on shorter evaluation horizons compared to the CARLA v2 leaderboard. While the CARLA v2 leaderboard uses long routes (7-10 kilometers) that are challenging to complete without errors, Bench2Drive employs shorter, more manageable scenarios1. This approach allows for a more granular assessment of driving skills and makes the benchmark more suitable for reinforcement learning applications.

**Checkpoints.** In particular, for VAD and UniAD, we directly use the checkpoint provide by Jia et al. (2024), which are trained on the whole Bench2drive dataset.

## F VISUALIZATION OF MODEL PREDICTION

In Figure 6, Figure 7 , Figure 8,Figure 9 and Figure 10, we show the comparative visualization of world model predictions across five different training settings (AdaWM, Alternate finetuning, model-only finetuning, policy-only finetuning, and no finetuning) reveals distinct performance patterns in the ROM03 task over 60 time steps. In each visualization set, the ego vehicle (red car) and surrounding agents (green cars) are shown with their respective planned trajectories (blue line for ego, yellow lines for others) across three rows: ground truth bird's-eye view (BEV), world model predicted BEV, and prediction error. All configurations exhibit increasing prediction errors as the time horizon extends further into the future, consistent with the growing uncertainty in long-term predictions. However, AdaWM demonstrates superior performance with notably smaller prediction errors compared to the other finetuning approaches, suggesting its enhanced capability in maintaining accurate world model predictions over extended time horizons.

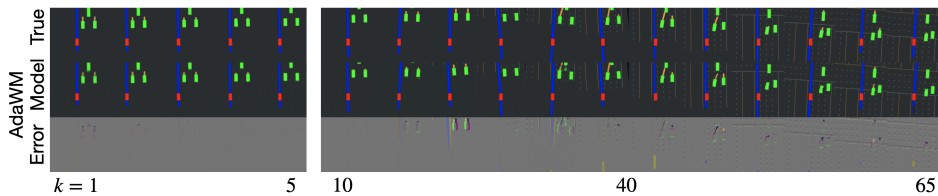

Figure 6: Prediction results for 65 time steps in AdaWM.

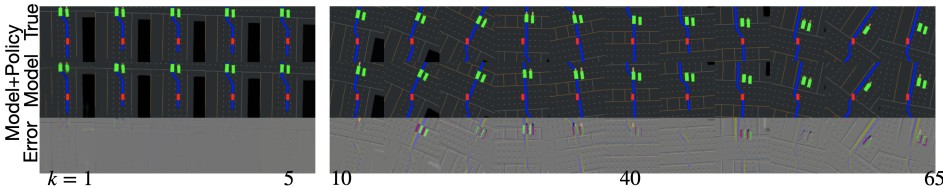

Figure 7: Prediction results for 65 time steps with alternate finetuning mechanism.

## G  SUPPLEMENTARY EXPERIMENTS

Furthermore, we conduct experiments on another five diverse and challenging scenarios to validate the performance of AdaWM. In particular, we consider the following environment settings, respetively,

- HI13: HighwayCutIn_Town13_Route23823_Weather7
- HE12: HighwayExit_Town12_Route2233_Weather9
- YE12: YieldToEmergencyVehicle_Town12_Route1809_Weather9
- BI12: BlockedIntersection_Town12_Route12494_Weather9
- VT11: VehicleTurningRoutePedestrian_Town11_Route27267_Weather12

HC13

```
HC13:    &HighwayCutIn_Town13_Route23823_Weather7
  env:
    world:
      town: Town13
      Weather: 7
      Route: 23823
    name: HC13
    observation.enabled: [camera, collision, birdeye_wpt]
    <<: *carla_wpt
    lane_start_point: [12.0, 7.0, 0.1, -90.0]
    ego_path: [[12.0, 7.0, 0.1], [23, 122, 0.1]]
    use_road_waypoints: [True, False]
    use_signal: False
```

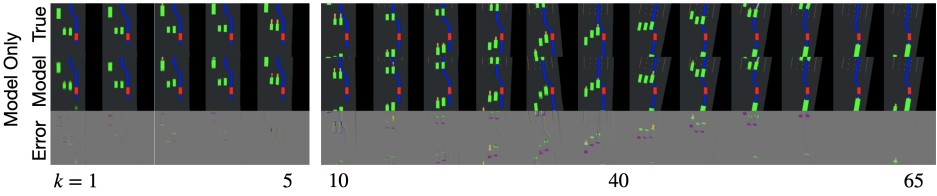

Figure 8: Prediction results for 65 time steps with only model finetuning.

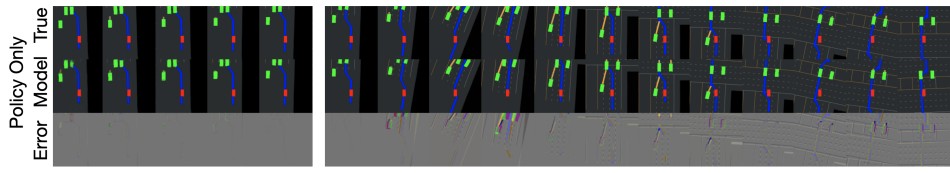

Figure 9: Prediction results for 65 time steps with only policy finetuning.

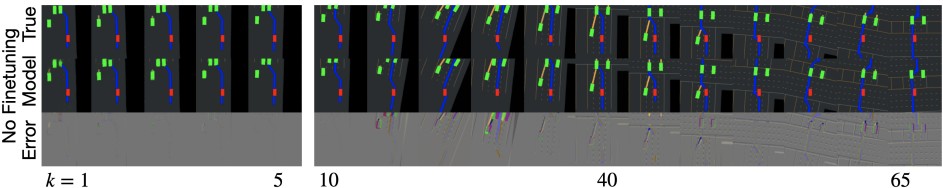

Figure 10: Prediction results for 65 time steps without finetuning.

## HE12

```
HE12:  &HighwayExit_Town12_Route2233_Weather9
  env:
    world:
      town: Town12
      Weather: 9
      Route: 2233
    name: HE12
    observation.enabled: [camera, collision, birdeye_wpt]
    <<: *carla_wpt
    lane_start_point: [32.0, -27.0, 0.1, -90.0]
    ego_path: [[32.0, -27.0, 0.1], [11, -129, 0.1]]
    use_road_waypoints: [True, False]
    use_signal: False
```

## YE12

```
YE12:  &YieldToEmergencyVehicle_Town12_Route1809_Weather9
  env:
    world:
      town: Town12
      Weather: 9
      Route: 1809
    name: YE12
    observation.enabled: [camera, collision, birdeye_wpt]
    <<: *carla_wpt
    lane_start_point: [11.0, 27.0, 0.1, -90.0]
    ego_path: [[11.0, 27.0, 0.1], [103, -21, 0.1]]
    use_road_waypoints: [True, False]
    use_signal: False
```

## BI12

```
BI12:  &BlockedIntersection_Town12_Route12494_Weather9
  env:
    world:
      town: Town12
      Weather: 9
      Route: 12494
    name: BI12
```

| | Pre-RTM03 | | HC13 | | HE12 | |
|---|---|---|---|---|---|---|
| Algorithm | TTC↑ | SR↑ | TTC↑ | SR↑ | TTC↑ | SR↑ |
| UniAD | 1.25 | 0.48 | 0.82 | 0.08 | 1.01 | 0.15 |
| VAD | 0.96 | 0.55 | 0.94 | 0.13 | 0.92 | 0.30 |
| DreamerV3 | 1.16 | 0.68 | 0.95 | 0.33 | 1.07 | 0.52 |
| **AdaWM** | 1.16 | 0.68 | **1.98** | **0.73** | **2.21** | **0.89** |

Table 7: The impact of finetuning in CARLA tasks. (HC/HE: highway cut-in / highway exit, 12 and 13 indicate different Towns.)

| | Pre-RTM03 | | YE12 | | BI12 | | VT11 | |
|---|---|---|---|---|---|---|---|---|
| Algorithm | TTC↑ | SR↑ | TTC↑ | SR↑ | TTC↑ | SR↑ | TTC↑ | SR↑ |
| UniAD | 1.25 | 0.48 | 0.22 | 0.04 | 0.42 | 0.10 | 0.24 | 0.05 |
| VAD | 0.96 | 0.55 | 0.79 | 0.08 | 0.88 | 0.18 | 0.82 | 0.12 |
| DreamerV3 | 1.16 | 0.68 | 0.80 | 0.15 | 0.93 | 0.42 | 1.02 | 0.58 |
| **AdaWM** | 1.16 | 0.68 | **1.53** | **0.52** | **1.41** | **0.59** | **1.21** | **0.88** |

Table 8: The impact of finetuning in CARLA tasks. (YE/BI/VT: Yield to emergency vehicle / Blocked Intersection / Vehicle Turing Route Pedestrian, 11 and 12 indicate different Towns.)

```
observation.enabled: [camera, collision, birdeye_wpt]
<<: *carla_wpt
lane_start_point: [80.0, 23.0, 0.1, -90.0]
ego_path: [[80.0, 23.0, 0.1], [121, 219, 0.1]]
use_road_waypoints: [True, False]
use_signal: False
```

VT11

```
VT11: &VehicleTurningRoutePedestrian_Town11_Route27267_Weather12
  env:
    world:
      town: Town11
      Weather: 12
      Route: 27267
    name: VT11
    observation.enabled: [camera, collision, birdeye_wpt]
    <<: *carla_wpt
    lane_start_point: [52.0, 98.0, 0.1, -90.0]
    ego_path: [[52.0, 98.0, 0.1], [-28, 76, 0.1]]
    use_road_waypoints: [True, False]
    use_signal: False
```

## G.1 EXPERIMENTS RESULTS

**Impact of Finetuning.** The experiments evaluate AdaWM across a diverse set of challenging autonomous driving scenarios, including highway maneuvers (cut-in and exit), emergency response (yielding to emergency vehicles), intersection navigation (blocked intersections), and complex interactions with pedestrians across different town environments and weather conditions. Looking at the impact of finetuning in Tables 7 and 8, AdaWM demonstrates substantial improvements over baseline methods (UniAD, VAD, and DreamerV3) across all scenarios. In highway scenarios (HC13 and HE12), AdaWM achieves remarkable gains, with the Success Rate (SR) improving from 0.33 to 0.73 in HC13 and from 0.52 to 0.89 in HE12, while Time-To-Collision (TTC) metrics show similar impressive improvements. The performance gains extend to more complex scenarios, with AdaWM showing particularly strong results in emergency vehicle response (YE12, SR from 0.15 to 0.52), blocked intersection handling (BI12, SR from 0.42 to 0.59), and vehicle-pedestrian interactions (VT11, SR from 0.58 to 0.88).

| Algorithm | HC13 | | HE12 | | YE12 | | BI12 | | VT11 | |
|---|---|---|---|---|---|---|---|---|---|---|
| | TTC ↑ | SR↑ | TTC ↑ | SR↑ | TTC ↑ | SR ↑ | TTC↑ | SR↑ | TTC↑ | SR↑ |
| No Finetuing | 0.95 | 0.33 | 1.07 | 0.52 | 0.80 | 0.15 | 0.93 | 0.42 | 1.02 | 0.58 |
| Policy-only | 1.32 | 0.52 | 1.92 | 0.72 | 1.33 | 0.44 | 1.30 | 0.49 | 1.09 | 0.76 |
| Model-only | **1.98** | 0.70 | 1.82 | 0.70 | 1.21 | 0.38 | 1.01 | 0.32 | 0.92 | 0.44 |
| Model+Policy | 1.68 | 0.60 | 1.94 | 0.80 | 1.42 | 0.47 | 1.21 | 0.42 | 1.01 | 0.70 |
| **AdaWM** | 1.92 | **0.74** | **2.21** | **0.89** | **1.53** | **0.52** | **1.41** | **0.59** | **1.21** | **0.88** |

Table 9: Comparison on the effectiveness of different finetuning strategies.

**Effectiveness of Finetuning Strategies.** The comparative analysis of different finetuning strategies in Table 9 demonstrates the superior effectiveness of AdaWM's alignment-driven approach. While conventional methods like policy-only, model-only, and alternating model+policy finetuning show some improvements over the baseline, AdaWM consistently outperforms them across all scenarios. For instance, in the highway exit scenario (HE12), AdaWM achieves an SR of 0.89 compared to 0.72 for policy-only and 0.70 for model-only approaches. This superior performance is particularly evident in complex scenarios like VT11, where AdaWM maintains high performance (SR 0.88) while other methods show significant degradation, with policy-only achieving 0.76 and model-only dropping to 0.44. These results demonstrate that AdaWM's integrated approach to alignment-driven finetuning is more effective than traditional methods at adapting to diverse driving scenarios while maintaining robust performance.

