# OpenReview forum: "AdaWM: Adaptive World Model based Planning for Autonomous Driving"
_ICLR.cc/2025/Conference — ICLR 2025 Poster_

### Official Review · Reviewer_4TZS · 2024-10-26

**Soundness:** 3
**Presentation:** 3
**Contribution:** 3
**Rating:** 6
**Confidence:** 4

**Summary:**

This paper introduces **AdaWM**, a World Model-based planning method that uses an efficient finetuning strategy to adapt to new tasks  different from the pre-training tasks.

The contributions of this paper are threefold:

1. The authors identify two main reasons for WM-based planning model performance degradation when adapting to new tasks: *dynamics model mismatch* and *policy mismatch*. They propose a novel method to quantify the severity of these mismatches.

2. An efficient finetuning strategy is proposed to adaptively update the dynamics model and policy based on the mismatch quantification.

3. The authors validate the effectiveness of the proposed AdaWM method using the CARLA simulator.

**Strengths:**

This paper introduce a novel method to quantify the two mismatches that leads to performance degradation. The mathematical derivation process is impressive.

**Weaknesses:**

The weaknesses of the paper are threefold:

1. ***The introduction and motivation of the paper lack clarity and conciseness.*** The authors allocate a significant portion of the introduction to discussing embodied AI and world models, which serve as the research background for this paper. It is advisable to provide a more concise overview, as excessive content on these topics detracts from the overall clarity and quality of the writing. For instance, the first two paragraphs could be shortened to two or three sentences. Additionally, the section titled **A Motivating Example** could also benefit from condensation.

2. ***The paper fails to adequately highlight the significance of its contributions and the necessity of the proposed method.*** After reviewing relevant studies, the authors should focus on clearly articulating the distinctions between their work and existing literature. Specifically, the authors should address the following questions, including but not limited to: (1) What distinguishes the proposed approach from existing methods reviewed? (2) Why can't the existing methods be directly applied to the tasks studied in this paper? (3) What challenges does this paper address that have not been solved by prior research?

3. ***The selection of baseline methods is not appropriate.*** It is not fair to compare the performance of a model finetuned for a specific task against the pretrained baseline models that have not undergone fine-tuning for that same task. Because finetuning on a specific task undoubtedly enhances the performance of the pre-trained model. Therefore, comparison against existing un-finetuned baselines (such as UniAD, VAD, DreamerV3) cannot adequately demonstrate the advantage of the proposed method. The authors should include existing fine-tuning methods (e.g. the alternate fine-tuning, model-only fine-tuning, and policy-only fine-tuning mentioned in the motivating example, as well as studies by Feng et al. (2023), Baker et al. (2022), Hansen et al. (2022), and Ouyang et al. (2022) reviewed in the related works section) as baselines for fairer comparison.

**Questions:**

Apart from the questions mentioned in the weaknesses, I have two more questions:

1. What is the rationale behind the authors' choice to employ the Low-Rank Adaptation (LoRA) algorithm for model fine-tuning, particularly considering that more advanced state-of-the-art (SOTA) fine-tuning methods (such as QLoRA) have already been developed? Does this mean the selection of finetuning algorithm does not affect the final performance? If so, it would be more convincing to include an ablation study to compare the performance when the proposed AdaWM is finetuned with different finetuning algorithms.

2. In the measurement of the **Mismatch of Policy**, why do the authors calculate the state distribution shift? In deep reinforcement learning (DRL), the distribution difference of a policy should be assessed based on the differences in actions, as the state serves merely as the input to the policy while the action is the output. It would be better to explain how the approach of using state distribution shift compares to the more standard approach of using action differences.

---

> ### Author Response · Authors · 2024-11-19
> **Reply to Reviewer 4TZS**
>
> ### Weakness
>
> We thank the reviewer for your suggestions and we address the raised concerns as follows.
>
> **A1. (Introduction)**  We appreciate the reviewer's feedback on the introduction's structure and conciseness. In the revision, we follow your suggestion to streamline the introduction by condensing the background on embodied AI and world models into a concise context-setting paragraph. We also refine the Motivating Example section to more directly connect to our technical contributions of  adaptive world models for autonomous driving.
>
>
> **A2. (Related work)**  We  restructure the related work discussion to explicitly highlight three key distinctions of our approach: (1) Unlike existing methods that rely on static models, AdaWM enables real-time adaptation to changing dynamics through our novel LoRA-based world model updates; (2) Current methods like UniAD and VAD cannot handle distribution shifts during deployment, while our approach continuously refines its understanding through online interaction; (3) AdaWM addresses the fundamental challenge of determining when and what to adapt during online interaction by introducing alignment-driven finetuning, which wasn't explored in previous research.
>
>
> **A3. (Baselines)** We appreciate the reviewer's comments about baseline comparisons. We want to clarify a crucial distinction in our experimental design: supervised learning methods like UniAD and VAD fundamentally cannot be fine-tuned during online interaction due to their reliance on ground-truth labels, which are unavailable during deployment. This limitation is precisely why we employ reinforcement learning, which makes continuous adaptation possible  through environmental interaction without requiring explicit supervision. We follow the standard comparison as conducted in [R1,R2] and provide the VAD and UniAD's performance  to demonstrate the advantages of online adaptation. For a fair evaluation of adaptation strategies, we do include comprehensive comparisons with relevant RL-based fine-tuning approaches, including model-only fine-tuning, policy-only fine-tuning, and alternate fine-tuning as ablation baselines (Table 2). These comparisons directly showcase the advantages of our selective adaptation mechanism over existing RL fine-tuning methods. We have provided a more detailed description on the comparison in our revision.
>
>
>
> - [R1] Li et al. "Think2drive: Efficient reinforcement learning by thinking in latent world model for quasi-realistic autonomous driving (in carla-v2).", 2024
>
> - [R2] Jia et al. "Bench2Drive: Towards Multi-Ability Benchmarking of Closed-Loop End-To-End Autonomous Driving", 2024
>
> ### Questions
>
> **A1. (Fintuning)** We appreciate the reviewer's question about fine-tuning algorithms. We want to clarify that AdaWM's core contribution lies in its adaptive mechanism for identifying and selectively updating either the world model or policy, rather than in the specific fine-tuning method used. Our approach is agnostic to the choice of parameter-efficient fine-tuning algorithm. We chose NoLa (a LoRa type of finetuning method, line 321 [R1]) for its simplicity and computational efficiency, but other methods like QLoRA could be readily integrated into our framework. This is because AdaWM's key innovation is in determining when and what to adapt based on mismatch detection, while the specific method of parameter updates is modular. Due to the scope of our study, we have focused on demonstrating the effectiveness of our selective adaption strategy rather than comparing different finetuning algorithms.
>
>
> **A2. (Policy Mismatch)** Thank you for raising this important point about policy mismatch measurement. We want to clarify that our state distribution metric actually measures the state visitation distribution, which is a direct consequence of policy behavior and it captures the states that the agent visits under its current policy. This is fundamentally different from treating states as mere inputs to the policy. In reinforcement learning, the state visitation distribution $\rho^{\pi}(x)=\sum\_{t}P(x=x\_t\vert \pi)$ is intrinsically linked to policy behavior as it reflects the long-term consequences of action selection. This approach provides a more comprehensive view of policy behavior than just comparing action distributions at individual states, as it captures the cumulative effect of the policy's decisions over time. This is particularly crucial in autonomous driving, where the sequence of visited states directly reflects the vehicle's trajectory and driving behavior.
>
> - [R1] Koohpayegani, Soroush Abbasi, et al. "NOLA: Compressing LoRA using Linear Combination of Random Basis." ICLR'24
>
>
> We hope that we clarified these very helpful comments. We would be glad to engage in discussion if there are any other questions or parts that need further clarifications.

---

> ### Comment · Reviewer_4TZS · 2024-11-21
> **Reviewer 4TZS's Reply to the Authors**
>
> I have carefully reviewed your responses and revisions, and I believe your work is valuable. However, the presentation of your method could lead to confusion and misinterpretation, which may undermine both the validity of your experiments and the perceived novelty of your approach. As a result, I have temporarily adjusted your score from 3 to 5.
>
> There are still two main points in your paper that require improvement in terms of writing:
>
> 1. *Logical Coherence*: I understand that you use UniAD and VAD to demonstrate the superiority of your method. However, as you mentioned, your core contribution is the adaptive fine-tuning strategy, AdaWM. There are at least two reasons why AdaWM might outperform all baseline methods: (1) Fine-tuned models may perform better on new tasks than those that have only undergone pre-training. (2) Adaptive fine-tuning may be more effective than other fine-tuning approaches. The first point has already been established by Julian et al. in [R1], and while you may reiterate this conclusion in the context of autonomous driving, you still need to demonstrate (2) to fully prove AdaWM's superiority. This is why I previously requested that you supplement your experiments with Table 2. I suggest clearly presenting these two points in the Experiment section and providing separate results to support them, which should help address concerns about the fairness of your baseline selection.
>
> 2. *Strength of Evidence*: In the Related Works section, you simply listed and summarized various papers and their contributions without highlighting their shortcomings in comparison to AdaWM. This omission could make it difficult for readers to grasp the significance of your proposed method. I recommend summarizing the deficiencies of existing methods based on the two points, (1) and (2), I mentioned earlier. Additionally, for the related works on RL fine-tuning or World Model fine-tuning, it would be helpful to explain why these methods are not included in your baselines (e.g., they may not be applicable to autonomous driving tasks) or to clarify how they fall into categories such as alternate fine-tuning, model-only fine-tuning, or policy-only fine-tuning.
>
> [R1] Ryan Julian, Benjamin Swanson, Gaurav Sukhatme, Sergey Levine, Chelsea Finn, and Karol Hausman. Never stop learning: The effectiveness of fine-tuning in robotic reinforcement learning. In Conference on Robot Learning, pp.2120–2136. PMLR, 2021.

---

> > ### Author Response · Authors · 2024-11-22
> > **Reply to Reviewer 4TZS**
> >
> > We sincerely thank the reviewer for engaging in the discussion promptly and providing the detailed and constructive feedback.
> >
> > We have made substantial revisions to address the two main concerns:
> >
> > 1. Logical Coherence.
> >
> > We have restructured our experimental analysis to clearly demonstrate AdaWM's superiority through two key aspects:
> > - Line 412 in Section 3.1 "*The Impact of Finetuning in Autonomous Driving*", where we compare AdaWM with baseline methods (supervised learning method and DreamerV3) to demonstrate the general benefits of fine-tuning
> > - Line 429 in Section 3.1, "*Effectiveness of Alignment-driven Fine-tuning Strategy*", where we compare AdaWM with other finetuning strategies used in previous work such as policy-only, model-only, model+policy with controlled experiments to show the effectiveness of AdaWM.
> >
> > Meanwhile, in Appendix G, we include five more experiments on very diverse scenarios in Bench2Drive (CARLA v2) dataset to verify 1) The impact of finetuning in Autonomous Driving Tasks (Table 7,8) and 2) Effectiveness of Alignment-driven finetuning (Table 9), i.e., highway cut in, highway exit, yield to emergency vehicle, blocked intersection, vehicle turning route pedestrian. By analyzing the obtained results, we obtain the consistent evaluation performance of AdaWM as in Section 3.
> >
> >
> > 2. Strength of Evidence
> >
> > We have thoroughly revised the Related Works section to:
> > - Organize existing methods into clear categories (supervised learning vs. world model-based approaches)
> > - Explicitly identify limitations of current approaches:
> >   * Supervised learning methods (VAD, UniAD) lack finetuning capabilities
> >   * World model methods focus primarily on pre-training
> >   * Existing fine-tuning approaches employ rigid, single-focused strategies (either model-only or policy-only)
> > - Added Table 1 to systematically compare related works across key dimensions: Learning Method, Fine-tuning Strategy, Online Interaction, and Tasks
> > We hope that these revisions have strengthened the paper's logical structure and provided clearer evidence for AdaWM's contributions.
> >
> > We hope our clarifications addressed the concerns effectively. We are happy to engage in further discussion if additional questions or points require further elaboration.

---

> ### Author Response · Authors · 2024-11-25
> **Thank you for your review**
>
> We greatly appreciate Reviewer's thorough review and constructive feedback. We hope that our response has addressed your concerns on our manuscripts and we are more than happy to answer any further questions if needed.

---

> > ### Comment · Reviewer_4TZS · 2024-11-26
> > **Reviewer 4TZS's Reply to the Authors**
> >
> > Thank you for your thorough revisions and your detailed response to my comments. I have reviewed your changes, and I appreciate how you have addressed all of my concerns. In particular, the modifications made to the related works section have significantly enhanced the clarity and the readability of the paper. Considering all aspects of the revisions, my final score will be 6 (marginally above the acceptance threshold).

---

### Official Review · Reviewer_kwJr · 2024-10-31

**Soundness:** 3
**Presentation:** 4
**Contribution:** 3
**Rating:** 6
**Confidence:** 3

**Summary:**

AdaWM is an adaptive world model-based planning approach for autonomous driving. It leverages pretrained dynamics models and policies in the finetuning setup to accelerate the learning process. This is an interesting approach to address performance degradation caused by distribution shifts in driving environments. AdaWM achieves state-of-the-art results in CARLA simulations, with its alignment-driven finetuning improving both efficiency and robustness.

**Strengths:**

AdaWM presents a novel world model approach to handle distribution shifts during finetuning, identifying mismatches in the dynamics model and policy as the main causes of performance drops. With its mismatch identification and alignment-driven finetuning, AdaWM mitigates performance degradation. Extensive experiments in CARLA with challenging tasks demonstrate AdaWM’s clear advantage in both success rate and time-to-collision over baseline models.

**Weaknesses:**

The paper could discuss how AdaWM adapts to different task complexities, like high-density urban areas versus highways, as these may impact its adaptability and performance. This leaves some concerns about AdaWM’s robustness across highly divergent scenarios.

Mismatch identification relies on total variation distance, which may add computational load in real time. It would be helpful to explore the complexity of this approach in more detail.

The LoRA-based low-rank adaptation for dynamics model finetuning is described as efficient, but its effect on prediction accuracy and model stability is unclear. How does this method balance efficiency with the accuracy needed for real-time decision-making?

Another potential concern is the generalizability and handling of corner cases. How does AdaWM handle extreme distribution shifts where the pretrained policies may not generalize well?

**Questions:**

Please refer to the questions in the weaknesses section.

---

> ### Author Response · Authors · 2024-11-19
> **Reply to Reviewer kwJr**
>
> We thank reviewer kwJr for your careful reading and valuation feedback. We address the raised concerns as follows.
>
> **A1. (Adaptability)** Our work follows previous works and chooses CARLA leaderboard v2 to evaluate the performance. In particular, leaderboard v2 contains large variety of scenarios including freeway and various densities in urban driving. Our experiments are conducted on routes that contains both urban intersections (ROM03, RTD12) and highway scenarios (LTM03, LTD03) and our results in Table 1-3 show consistent performance improvement across these varying complexities.
>
> **A2. (TV Distance)** We clarify that in our implementation, the total variation distance computation is lightweight, requiring only element-wise operations on probability distributions and our implementation optimizes this calculation through vectorization. The computational overhead is negligible ($<1$ms) compared to the planning cycle ($100ms$).
>
> **A3. (Trade-off)** We thank the reviewer for pointing out the tradeoff between accuracy and efficiency, especially in the online planning tasks. In our work, AdaWM achieves such balance by following two designs: 1) We adopt NoLa (a LoRa type of finetuning method, line 321 [R1]) which decouples the number of trainable parameters from both the choice of rank and the network architecture. This flexibility allows for fine-tuning with fewer parameters while maintaining accuracy. 2) AdaWM conduct finetuning based on performance feedback. Specifically, at each step, AdaWM estimates the mismatch of model and policy and determines  the dominate root cause for the performance degradation and hence guide the finetuning to reduce the gap.
>
>
> **A4. (Testing Scenarios)**  We thank the reviewer for raising the corner cases and we clarify that the testing scenarios are from CARLA leaderboard v2 tasks (Bench2drive dataset), which focus on the corner cases. For instance, tasks  RTD12 requires the ego vehicle is performing an unprotected left turn at an intersection, yielding to oncoming traffic. In the case when the extreme distribution shifts happen and the pre-trained polices do not generalize well (such that from the case with traffic signal to the case of without traffic signal), AdaWM is capable of identify the root cause first (through mismatch identification) followed by alignment-driven finetuning (ref. the comparison of performance RTM03 and LTD03 in Table 1).
>
>
> - [R1] Koohpayegani, Soroush Abbasi, et al. "NOLA: Compressing LoRA using Linear Combination of Random Basis." ICLR'24
>
> We hope that we clarified these very helpful comments. We would be glad to engage in discussion if there are any other questions or parts that need further clarifications.
>
> ---
> *Update in the revision*: We thank the reviewer for their valuable advice on improving our work. In the revised Appendix G, we include five additional experiments across diverse scenarios in the Bench2Drive (CARLA v2) dataset, including two *freeway scenarios*: highway cut-in, highway exit, yielding to emergency vehicles, blocked intersections, and vehicle-turning pedestrian routes. The results (Tables 7–9) further validate the effectiveness of alignment-driven fine-tuning, demonstrating consistent evaluation performance for AdaWM as presented in Section 3.

---

> > ### Comment · Reviewer_kwJr · 2024-11-25
> >
> > Thanks for your detailed response. I will keep my positive recommendation.

---

### Official Review · Reviewer_UVY6 · 2024-11-02

**Soundness:** 3
**Presentation:** 3
**Contribution:** 2
**Rating:** 5
**Confidence:** 3

**Summary:**

AdaWM introduces an adaptive world model-based planning method that selectively fine-tunes the policy and model to mitigate performance degradation. Theoretical analyses are provided, and experiments are conducted on well-known CARLA benchmarks.

**Strengths:**

- This paper makes it easy for readers to grasp the main idea.
- This paper includes theoretical analysis.

**Weaknesses:**

- The author needs to provide visualizations of the model's prediction quality and trajectory planning quality to further demonstrate the model's performance.
- The author should compare more model-based/world model planning methods in autonomous driving. Because UniAD and VAD are end-to-end methods, and Dreamer is not specifically designed for autonomous driving setting.
- The method requires model and policy rollouts, which may introduce significant safety issues in the real world. Although this work addresses this by using simulation, the simulation domain gap may lead to out-of-distribution problems in real-world applications.

**Questions:**

- What is the reason the author chose different fine-tuning methods for the policy and model?
- Is the BEV used by the author constructed from ground truth perception labels, or is it built directly using image inputs?
- Are the baselines trained on the ROM03, RTD12, LTM03, and LTD03 tasks?

---

> ### Author Response · Authors · 2024-11-19
> **Reply to Reviewer UVY6**
>
> ### Weakness
>
> **A1. (Prediction Quality)** We thank the reviewer for your comments and we update the visualizations of the model's prediction quality in Appendix F. Meanwhile, we provide the planning visualization (in gif) in the supplementary files.
>
> **A2. (Baselines)** We appreciate the reviewer's suggestion regarding comparative methods. Our work specifically examines real-time adaptation for world-model based approach, where Think2Drive [R1] (based on DreamerV3) currently holds state-of-the-art performance on the CARLA leaderboard v2 (Bench2Drive [R2]), making it the most relevant baseline method. Meanwhile, we follow the well-established works [R1,R2] and include the results on UniAD and VAD for comparison and we use the checkpoints provided by Bench2Drive for those methods for a fair comparison.
>
> More importantly, both methods use fundamentally different methodological approaches and such comparison aims to show the advantages of online adaptation in RL-based methods against traditional supervised learning paradigms (which can not adapt during online interaction when  the label is not available). This comparative framework allows us to demonstrate the distinct advantages of adaptive planning while maintaining scientific rigor by benchmarking against current top-performing systems.
>
> **A3. (Online Adaptation)** In response to the reviewer's concern, in the following we elaborate on  the distinction between simulation and learned world models in our approach.  Our method's world model is learned through interactions with physical engine (specifically CARLA) and represents the agent's internal understanding of dynamics, which is similar to how humans develop mental models through experience. Moreover,  the learned world model is different from the simulation and can adapt to new scenarios and handle out-of-distribution issues through online updates, making it fundamentally different from using  fixed simulation-based approaches. More importantly, the world model's adaptive nature actually provides an advantage for handling domain gaps since it can continuously refine its predictions and adapt to distribution shifts based on real-world observations, unlike static policies that rely solely on training data. This online adaptation capability is particularly crucial for bridging any simulation-to-real gaps that may emerge during deployment.
>
>
> ### Questions
> **A1. (Finetuning Method)**  The different adaptation strategies are chosen based on their specific roles. For instance, the learned world model contains core knowledge on the latent representation and should be preserved during finetuning. To this end, the model parameters are decomposed into two lower-dimensional vectors: the latent representation base vector and weight vector. LoRa is suitable for such product structure and can enable efficient updates of dynamics predictions while preserving  such core knowledge. Meanwhile, the convex combination approach for policy allows rapid behavior adaptation through reweighting existing skills rather than learning entirely new ones. We thank the reviewer for your suggestions and we have clarified the rationales for our choice in the revision (\textbf{Section 2.2})
>
>
> **A2. (BEV)**  We clarify we follow the standard approach as in [R1,R2,R3,R4] (including the bench2drive dataset) to use privileged BEV instead of a BEV former, which is beneficial evaluate the core decision-making capabilities of our planning algorithm.
>
>
>  **A3. (Training Tasks)** UniAD and VAD baseline are trained on open source Bench2Drive dataset [R1,R2], where all four tasks, ROM03, RTD12, LTM03 and LTD03 are included.
>
>
>
> - [R1] Li et al. "Think2drive: Efficient reinforcement learning by thinking in latent world model for quasi-realistic autonomous driving (in carla-v2).", 2024
>
> - [R2] Jia et al. "Bench2Drive: Towards Multi-Ability Benchmarking of Closed-Loop End-To-End Autonomous Driving", 2024
>
> - [R3] Chen et al., "Learning to drive from a world on rails." ICCV. 2021.
>
> - [R4] Gao et al. "Enhance sample efficiency and robustness of end-to-end urban autonomous driving via semantic masked world model." IEEE Transactions on Intelligent Transportation Systems, 2024.
>
>
> We hope that we clarified these very helpful comments. We would be glad to engage in discussion if there are any other questions or parts that need further clarifications.

---

> > ### Comment · Reviewer_UVY6 · 2024-11-23
> >
> > Thank you for the author's response. However, some questions remain:
> >
> > - Thank you for providing the visualization of the model’s performance, but I noticed that the vehicle's driving behavior is not smooth. In other words, the trajectory is not comfortable, which makes it challenging to apply in real autonomous driving scenarios. Was this issue considered in the evaluation?
> > - The author mentioned: "More importantly, the world model's adaptive nature actually provides an advantage for handling domain gaps since it can continuously refine its predictions and adapt to distribution shifts based on real-world observations, unlike static policies that rely solely on training data." The author needs to provide more evidence to avoid overclaiming. As I understand, if the domain gap between simulation (e.g., CARLA) and real-world scenarios is too large, the trained world model might require retraining from scratch. How can the proposed method ensure better transferability?
> > - For the comparison with UniAD and VAD, are they still using image-based inputs? Pure image inputs and the BEV inputs used by the author seem inherently incomparable. Could the author provide a more detailed explanation?

---

> ### Author Response · Authors · 2024-11-24
> **Reply to Reviewer UVY6 (1/3)**
>
> We sincerely thank the reviewer for engaging in the discussion and providing us  the detailed feedback. We address the raised concerns as follows.
>
> **A1 Driving Smoothness**: We appreciate the reviewer's attention to driving smoothness and checking our supplementary material.
>
> **Reward Structure**: Our reward function comprehensively includes  *multiple critical factors* and the smoothness is not the only factors to consider in terms of the driving performance. We follow the standard approach [R1,R2] and consider six factors: driving smoothness, safety (collision or not), target reaching, velocity, orientation, and traveling time. While we fully agree that smoothness is important for passenger comfort, autonomous driving requires careful balancing of all these factors simultaneously.
>
> **Navigation in the Challenging CARLA Scenarios**: The trained agents must learn to navigate complex trade-offs in real-world scenarios. An agent could achieve perfect smoothness by driving very slowly, but this would severely compromise travel time and target reaching objectives. In particularly challenging scenarios like unprotected right turns in dense traffic (one of our CARLA leaderboard v2 test cases), the agent must sometimes make decisive movements when safe gaps appear, requiring *temporary trade-offs between smoothness and successful task completion*.
>
> Our evaluation specifically focuses on highly challenging scenarios from the CARLA leaderboard v2, including complex situations like unprotected right turns with dense traffic and yielding to emergency vehicles. The observed trajectories reflect our agent learning to balance these competing objectives in these demanding scenarios, where perfect smoothness might need to be temporarily compromised to achieve critical safety and task completion goals.
>
>
> - [R1] Knox, W. Bradley, et al. "Reward (mis) design for autonomous driving." Artificial Intelligence 316 (2023): 103829.
>
> - [R2] Kiran, B. Ravi, et al. "Deep reinforcement learning for autonomous driving: A survey." IEEE Transactions on Intelligent Transportation Systems 23.6 (2021): 4909-4926.

---

> > ### Author Response · Authors · 2024-11-24
> > **Reply to reviewer UYV6 (2/3)**
> >
> > **A2 Transferability**: We sincerely thank the reviewer for this important question about domain adaptation claims.
> >
> >
> > We first clarify our main contribution is making the finetuning process significantly more efficient through alignment-driven adaptation.
> >
> > - **Comparing with other finetuning method**. As demonstrated in Table 3 and Table 9, AdaWM achieves substantially better performance after just 1 hour of finetuning in new tasks compared to conventional approaches (policy-only, model-only, and alternate finetuning). This efficiency is particularly valuable for real-world deployment scenarios where adaptation time is limited.
> > - Moreover, the **supervised learning approach**, on the other hand, requires labeled data for training (and finetuning). This is the fundamental difference from RL, which relies on "real-world observations" to "continuously refine its (world model's) predictions (through online interaction)" as in our original statement. We also provide Table 1 in our revision for the comparison of our work and previous work in order to highlight our contribution.
> >
> > The enhanced transferability of AdaWM is achieved through several innovative technical components working in concert.
> >
> > - **Key Design for efficiency.** We decompose model parameters into a latent representation base vector and a weight vector. The base vector captures fundamental *temporal and spatial relationships* that are likely remain constant across tasks (as a key benefits of using a latent representation [R1,R5,R6]), while only the weight vector is adapted during finetuning. Rather than learning entirely new behaviors, our policy uses a convex combination approach to rapidly adapt by reweighting *existing skills*. Furthermore, our alignment-driven finetuning first identifies key performance mismatches and focuses adaptation resources there, avoiding wasted effort on less critical factors.
> > - **Experiments on various challenging scenarios.** We provide extensive validation through Section 3. In our revision Appendix G, we also provide supplementary experiments, where we test AdaWM in five challenging scenarios completely unseen during pre-training: highway cut-in maneuvers, highway exit navigation, emergency vehicle yielding, blocked intersection handling, and vehicle-turning with pedestrian interaction. The results demonstrate that after just 1 hour of finetuning, AdaWM achieves superior performance compared to alternative finetuning methods in these novel scenarios, providing strong evidence for our method's efficient adaptation capabilities. We note that CARLA leaderboad tasks are well-known to be highly challenging. Our work focuses on the  CARLA leaderboard V2, a well known hard benchmark for end-to-end algorithm. In fact, the current SOTA method [R3,R4] still struggle with solving all tasks after initial training.
> >
> >
> > - [R3] Li et al. "Think2drive: Efficient reinforcement learning by thinking in latent world model for quasi-realistic autonomous driving (in carla-v2).", 2024
> >
> > - [R4] Jia et al. "Bench2Drive: Towards Multi-Ability Benchmarking of Closed-Loop End-To-End Autonomous Driving", 2024
> >
> > - [R5] Pak et al. "CarNet: A dynamic autoencoder for learning latent dynamics in autonomous driving tasks." 2022.
> > - [R6] Chen et al. "End-to-end autonomous driving perception with sequential latent representation learning." 2020 IROS

---

> > > ### Author Response · Authors · 2024-11-24
> > > **Reply to reviewer UYV6 (3/3)**
> > >
> > > **A3 BEV Inputs**: Thank you for this important question about input modality comparisons.
> > >
> > > We acknowledge that UniAD and VAD are supervised learning methods and learn **hidden representation of BEV** (both methods can not directly use privileged BEV as input). For fair comparison, both UniAD and VAD are tested using checkpoint provided by Bench2Drvie dataset [R1,R2].
> > >
> > > We clarify we follow the standard approach as in [R1,R2,R3,R4] (including the bench2drive dataset) to use privileged BEV as input, which is beneficial to evaluate the core decision-making capabilities of our planning algorithm. The main scope of our study is on advancing planning robustness under distributional shifts, indicating that the perception component is only tangential to our primary research objectives. Meanwhile,  as  evidenced by our comparison with state-of-the-art method Think2Drive, which also use a privileged BEV for their analyses and comparisons,  their method suffers from performance degradation despite using privileged BEV.  Hence, using a privileged BEV enables a more rigorous evaluation of our core technical contributions in planning adaptation, as it prevents perception errors from obscuring the performance differences in planning strategies.
> > >
> > > - [R1] Li et al. ``Think2drive: Efficient reinforcement learning by thinking in latent world model for quasi-realistic autonomous driving (in carla-v2).", 2024
> > >
> > > - [R2] Jia et al. ``Bench2Drive: Towards Multi-Ability Benchmarking of Closed-Loop End-To-End Autonomous Driving", 2024
> > >
> > > - [R3] Chen et al., "Learning to drive from a world on rails." ICCV. 2021.
> > >
> > > - [R4] Gao et al. "Enhance sample efficiency and robustness of end-to-end urban autonomous driving via semantic masked world model." IEEE Transactions on Intelligent Transportation Systems, 2024.
> > >
> > >
> > > We hope our clarifications addressed the concerns effectively. We are happy to engage in further discussion if additional questions or points require further elaboration.

---

> > > > ### Comment · Reviewer_UVY6 · 2024-11-25
> > > >
> > > > Thank you for the detailed reply.
> > > > - Regarding the algorithm's visualization not being smooth, I noticed that the vehicle also exhibits jittering on an empty straight road before turning. I acknowledge that the design of RL rewards requires balancing various metrics, but for performance in simple scenarios, this is not a policy that can be directly deployed.
> > > > - Concerning transferability, the author responded about transferring between different scenarios within the CARLA environment. From a practical standpoint, it raises the question of how efficiently a model trained in a simulator can be transferred to real-world settings.
> > > > - Lastly, regarding the baseline, since different types of inputs with varying difficulties are used, comparing the two does not adequately demonstrate the algorithm's advantages. The author emphasized focusing on the model's planning capabilities; a better choice would be to compare with more models that use data processed post-detection for planning, such as the nuPlan benchmark, otherwise, it is difficult to fully demonstrate the model's advantages.
> > > >
> > > > Overall, I maintain my score.

---

> ### Author Response · Authors · 2024-11-26
> **Thank you for your follow up response**
>
> We thank the reviewer for engaging in the discussion. Below are our detailed responses to each concern:
>
> **A1. Smoothness.**
>
> While our primary focus is on handling challenging CARLA scenarios (e.g., long tail events such as unprotected traffic, pedestrian on the driveway.), we acknowledge the importance of smooth motion in basic driving conditions.
>
> We note that this issue is common when testing in the challenging scenarios, in which other baseline methods are performing even less smooth behaviors comparing with AdaWM. This qualitatively shows that AdaWM has the capability of reducing the jittering. We have included additional qualitative experiments results (in .gif) in simple driving scenarios. The supplementary materials now include new driving records demonstrating smooth performance in basic conditions (ref. Stable Performance Demo/adawm-right\_turn\_spectator.gif,adwm-going straight,Dreamerv3\_RightTurn.gif)
>
> In particular, for the simple scenarios, our framework allows for adaptive behavior refinement through continued training in simpler environments, enabling the system to effectively optimize its performance to obtain "smooth behaviors".
>
> **A2. Transferability**
>
> We acknowledge that bridging the gap in sim-to-real is an important direction and highly nontrivial, and there are separate line of works on addressing this issue [R1], which is beyond the scope of our work. Our work's scope is to provide  insights into handling complex driving scenarios that are rather essential for autonomous driving. To this end, we choose CARLA as our testing bed, which is the current state-of-the-art benchmark for autonomous driving algorithms due to its high-fidelity physics simulation that closely mirrors real-world dynamics. The environment provides realistic sensor models and environmental conditions. Our approach follows well-established validation protocols from previous works in the field and demonstrate the performance of our proposed AdaWM.
>
> - [R1] Yang et al. "Drivearena: A closed-loop generative simulation platform for autonomous driving." arXiv preprint arXiv:2408.00415 (2024).
>
> **A3. Benchmark beyond CARLA**
>
> We thank the reviewer for suggesting nuPlan as a comparison benchmark. We would like to clarify our choice of CARLA over nuPlan [R2] was aligned with our research objectives.
>
>  We follow the SOTA method Think2drive and Bench2drive on close-loop autonomous driving to set-up our experiments on CARLA leaderboard v2 tasks. CARLA enables end-to-end evaluation of direct sensor processing and provides comprehensive testing of *interactive behaviors* in complex scenarios. It is specifically designed to evaluate *long-tail events* and serves as the primary benchmark for end-to-end autonomous driving systems.
>
> - [R2] Hallgarten, Marcel, et al. "Can Vehicle Motion Planning Generalize to Realistic Long-tail Scenarios?." arXiv preprint arXiv:2404.07569 (2024).
> ---
> We hope our responses have sufficiently addressed the issues raised. If there are any remaining concerns or areas requiring further clarification, please don’t hesitate to let us know. We would be happy to provide additional explanations or make further adjustments as needed. Thank you again for your time and effort.

---

> ### Author Response · Authors · 2024-12-01
> **A Gentle Reminder**
>
> Dear Reviewer,
>
> As the rebuttal period is nearing its conclusion, we would like to kindly follow up to see if there are any additional questions or areas that need further clarification. Thank you once again for your time and valuable feedback.
>
> Authors

---

> > ### Comment · Reviewer_UVY6 · 2024-12-02
> >
> > Thank you for the author's response.
> >
> > Regarding the smoothness issue, as I understand it, many imitation learning algorithms are capable of generating smooth trajectory outputs. I think the jitter might be a result of using reinforcement learning algorithms, but it seems that fine-tuning the policy hasn’t fully addressed this issue. Additionally, I still believe that the baselines selected by the author, particularly UniAd and VAD, may not be the most suitable for this setting. Under this circumstances, I think there is a lack of more comprehensive experimental comparisons.
> >
> > So, regretfully, I cannot raise my score.

---

> > > ### Author Response · Authors · 2024-12-03
> > >
> > > We appreciate the reviewer's continued engagement with our work. We would like to further clarify the concerns regarding trajectory smoothness and baseline selection with additional context and clarification.
> > >
> > > **A1 (Challenging Tasks).**  The perceived jittering in trajectories is primarily a reflection of the **inherent complexity of the CARLA Leaderboard V2 tasks** (Bench2Drive dataset [R2]) rather than a limitation of our approach. These scenarios present extreme challenges that require rapid, adaptive decision-making in response to dynamic obstacles and complex road conditions. The difficulty of these scenarios is well-documented and methods like UniAD and VAD achieve **very low success rates or fail completely** on these challenging tasks, as demonstrated in [R1] and our results in Table 1-4. Our method makes a deliberate balance on task completion, safety and trajectory smoothness through the reward structure.
> > >
> > > This prioritization has proven effective, as evidenced by our superior Success Rate (SR) and Time-to-Collision (TTC) metrics across all test scenarios (Tables 1-7). While trajectory smoothness was not our main optimization target, our comparative analysis shows that our approach actually helps mitigate jittering effects when compared to other fine-tuning methods and the current SOTA method Think2Drive. This improvement in smoothness comes as an additional benefit on top of our main contributions in adaptation and safety. We will include the discussions  on the smoothness and jittering in the discussion section of our revision.
> > >
> > > **A2 (Baseline).** We clarify the experiments design and baseline selection as follows. Our work specifically examines real-time adaptation for world-model based approach. To this end, as also suggested by *Reviewer 4TZS*, we first show "*The Impact of Finetuning in Autonomous Driving*" to demonstrate the necessity of the finetuning for autonomous driving applications by comparing with two representative imitation learning based methods UniAD and VAD. Both methods can not be finetuned during the online interaction (since the label is not available). Our results demonstrate their poor performance with low SR and low TTC.
> > >
> > > Next, we validate the *efficiency of finetuning strategy* with state-of-the-art  RL based method Think2Drive [R1] (based on DreamerV3) that  equipped with model-only fine-tuning, policy-only fine-tuning, and alternate fine-tuning strategies.
> > >
> > > Our work has this two-pronged experimental design, comparing against both non-adaptive baselines (VAD, UniAD) and various fine-tuning strategies over 9 challenging tasks in CARLA leaderboard v2, provides evidence for both the necessity of online adaptation and the superiority of our alignment-driven adaptation mechanism.
> > >
> > > - [R1] Li et al. "Think2drive: Efficient reinforcement learning by thinking in latent world model for quasi-realistic autonomous driving (in carla-v2).", 2024
> > >
> > > - [R2] Jia et al. "Bench2Drive: Towards Multi-Ability Benchmarking of Closed-Loop End-To-End Autonomous Driving", 2024
> > >
> > >
> > > We would like to thank the reviewer again for his valuable feedback and discussions, and hope that our clarifications have adequately addressed the concerns.

---

### Official Review · Reviewer_EsQt · 2024-11-03

**Soundness:** 3
**Presentation:** 2
**Contribution:** 2
**Rating:** 6
**Confidence:** 4

**Summary:**

The authors proposed a model-based planning method, AdaWM, to overcome the performance degradation between the offline pre-trained model and the online finetuned model. The authors first investigated the root of this issue, which are the distribution shift of both planning policy and dynamic model. Based on that, their proposed method AdaWM follows two steps for finetuning, by first identifying the the cause of the mismatch and then selectively updating the mismatched part via LoRA. The experiments are conducted on challenging Carla benchmark to demonstrate the effectiveness of the proposed method.

**Strengths:**

1. A motivating point-of-entry for analyzing distributional shift during adaptation in model-based planning for autonomous driving. Quantified by the distributional gap of dynamic model or planning policy, an adaptive finetuning framework (AdaWM) is proposed that selectively fine-tunes world model or planning policy given switch-based dominated mismatch assessment.
2. Analytical characterization and assessment of the performance gap for world model and planning policy.
3. Improved adaptation results by AdaWM finetuning under several CARLA scenarios.

**Weaknesses:**

1. Unverified model configurations and experimental settings:
*  The detailed setup for both the world model (dynamic) and the pretrained planning policy in AdaWM is unclear.
*  The paper lacks comprehensive configurations and descriptions of the experimental scenarios in CARLA.
*  The metrics selected are not entirely objective, as key benchmarks such as Driving Score and Route Completion are missing. Additionally, using a self-defined reward cannot ensure objective measurement of performance degradation.
2. Concerns regarding objectivity and completeness in experimental validation:
* Lack of benchmarked results for AdaWM in CARLA, such as on Bench2Drive or CARLA Leaderboard-2.
* Potentially unfair comparisons with other baselines, as AdaWM utilizes perfect BEV perception as input, whereas other baselines (e.g., VAD, UniAD) employ end-to-end pipelines with raw sensor inputs.
* Insufficient evaluation of generality in AdaWM's ablation studies. As a general pipeline for finetuning both the world model and planning policy, the results using various state-of-the-art models (e.g., VAD v1-v2, UniAD, TCP, PlanT) remain unclear.

**Questions:**

In addition to the following concerns that need to be addressed, several questions remain regarding the methodology and experiments:
1. Is the reward curve objective enough to reflect performance degradation? What if the reward is poorly crafted, or the adapted scenarios introduce a distributional shift in the reward? Further evaluation with metrics like driving score or success rate is recommended.
2.  How are \( C_1 \) and \( C_2 \) formulated in relation to Theorem 1? Additional clarification on the derivation would be helpful.
3. What is the purpose of introducing the switching strategy? How does it affect performance (e.g., driving score or the respective performance gap for the world model and policy) when both structures are adaptively fine-tuned?
4. How well does AdaWM perform in fine-tuning current well-known policies within the community?

---

> ### Author Response · Authors · 2024-11-19
> **Reply to Reviewer EsQt (1/3)**
>
> ### Weakness
>
> We thank the reviewer EsQt for your careful reading and valuable comments. We provide further clarifications below.
>
> **A1.** (1) **Pretraining details**.   The world model and the planning policy are pretrained based on Dreamer v3 (Hafner et al. 2023). The training details  can be found in Appendix E in our revision, and we also outline the key steps as follows:
>
> - World Model Pretraining: As stated in line 195 and 1008, the world model is implemented as a Recurrent State-Space Model (RSSM) to learn the environment dynamics, encoder, reward, continuity and encoder-decoder. This is achieved by optimizing the representation loss, dynamics loss and prediction loss.
> - Policy Pretraining: The policy is derived by using actor-critic with 16 steps look-ahead. As stated in line 393, the pre-training is conducted on 1 single V100 for 12 hours.
>
>
> (2) **Tasks configuration**. We include the details of the task configuration in Appendix D (line 938). The tasks considered in our work use the same scenarios defined in CARLA leaderboard V2 and Bench2drive. In particular, we use the starting point and the end point appointed by the CARLA leaderboard 2.0 benchmark. In our revision, we provide all the tasks' configurations details in \textbf{Appendix D }(line 938).
>
>
> (3) **Metrics**. We first clarify that the success rate (SR) used in our comparison is the same as the completion rate, i.e., percentage of the trails that do not have any failure incidents. In our revision, we also updated the terminology to avoid confusion (ref. line 402).  We also appreciate the reviewer mentioning the driving score metric in CARLA benchmark. However, we choose not to use Driving score (DS) due to its limitations on evaluating the algorithm performance ( see [R1,R2] below). Specifically,  DS is known to have high variance due to its multiplicative nature, where a single infractions
>  disproportionately impact the overall performance evaluation. This high variance makes it less reliable for comparing algorithmic improvements. Meanwhile, we acknowledge Weighted Driving Score (WDS) attempts to address some of these issues, it also has limitations, such as a hand-crafted weighting schemes may not reflect real-world priorities.
>
> For instance, the definition of WDS is $\mathrm{WDS}=\mathrm{RC} * \prod\_i^m \text{ penalty }\_i^{n\_i}$, where RC means route completion rate and penalty and $n_i$ is determined by the number of infractions and scenario density, and these factors are tied to the testing dataset and may not accurately reflect real-world conditions. In our work, we choose TTC, a critical metrics for evaluating human driving safety. In our experiments, we use success rate (SR) to evaluate the overall performance, while TTC reflects safety-critical aspects of driving behavior.
>
>  (4) **Reward**. We emphasize that reward serves as a learning signal and it is a standard way to use the accumulated reward to evaluate the *learning performance*.  We also note that SOTA approaches ([R1, R2]) on this scope also use same accumulated reward for their model learning. For instance, in [R1], the reward is defined as $r=r\_{\text {speed }}+\alpha\_{\text{tr}} r\_{\text {travel }}+\alpha\_{\text {de }} p\_{\text {deviation }}+\alpha\_{\text {st}} c\_{\text{steer}}$. We follow those well-established works and use the reward to incorporate many factors including safety (relevant to TTC) and target reaching (relevant to success rate).
>
>  Moreover, we *evaluate* the final driving performance using widely accepted metrics like TTC, which are distinct from the training reward.   As expected, the *learning performance* is directly related to the *driving performance*, and  it can be seen in Tables 1,2,3, (e.g., Dreamer V3's performance) the learning performance is directly related to the driving performance and the driving performance indeed can suffer from performance degradation when the agent is facing a new scenario.
>
>  - [R1] Li et al. "Think2drive: Efficient reinforcement learning by thinking in latent world model for quasi-realistic autonomous driving (in carla-v2).", 2024
>
> - [R2] Jia et al. "Bench2Drive: Towards Multi-Ability Benchmarking of Closed-Loop End-To-End Autonomous Driving", 2024

---

> ### Author Response · Authors · 2024-11-19
> **Reply to Reviewer EsQt (2/3)**
>
> **A2.** (1) **Benchmark results**. We would like to point out that AdaWM is indeed conducted on the scenarios defined in learderboard v2 and Bench2Drive (Appendix E). For comparison, we include the results reported by Bench2Drive in our revision.
>
> (2) **BEV**. We clarify we follow the standard approach as in [R1,R2,R3,R4] (including the bench2drive dataset) to use privileged BEV instead of a BEV former, which is beneficial evaluate the core decision-making capabilities of our planning algorithm. While we acknowledge that UniAD and VAD tackle the full autonomous driving stack including perception, both methods do not accept ground truth BEV as it learns hidden representations for BEV rather than using ground truth BEV. Moreover, the main scope of our study is on advancing planning robustness under distributional shifts, indicating that the perception component is only tangential to our primary research objectives. Meanwhile,  as  evidenced by our comparison with state-of-the-art method Think2Drive, which also use a privileged BEV for their analyses and comparisons,  their method suffers from performance degradation despite using privileged BEV.  Hence, using a privileged BEV enables a more rigorous evaluation of our core technical contributions in planning adaptation, as it prevents perception errors from obscuring the performance differences in planning strategies.
>
>
> - [R3] Chen et al., "Learning to drive from a world on rails." ICCV. 2021.
>
> - [R4] Gao et al. "Enhance sample efficiency and robustness of end-to-end urban autonomous driving via semantic masked world model." IEEE Transactions on Intelligent Transportation Systems, 2024.
>
> (3) **Evaluation of AdaWM**. We first clarify that AdaWM is a *world model based approach* for planning, while  addressing the performance degradation issue during *online interaction*. We acknowledged that it is feasible to use our proposed framework for finetuning VAD and UniAD, but non-trivial treatments are needed to modify the original methods' structure, as outlined as follows:
>
> -  VAD and UniAD are designed as end-to-end networks that directly map sensor inputs to driving actions through supervised learning, without explicitly modeling environmental dynamics, not mentioning the latent dynamics model. This world model is essential for our reinforcement learning-based adaptation, as it enables counterfactual reasoning about different actions and their consequences, allowing the system to learn from simulated experiences during online adaptation.
>
> - More importantly,  the supervised learning objectives used in VAD and UniAD are also inherently different from RL approach, where they optimize for prediction accuracy against human demonstrations (i.e., label is needed), whereas our method optimizes for expected future returns through environment interaction (i.e., no label is available).
>
> - Moreover, while theoretically one could attempt to retrofit VAD or UniAD with a world model (such as  through knowledge distillation), this would require fundamental architectural changes that would essentially transform them into different methods altogether.

---

> ### Author Response · Authors · 2024-11-19
> **Reply to Reviewer EsQt (3/3)**
>
> ## Questions
>
>
> **A1. (Reward)**  We first clarify that reward is used for signaling the learning performance by considering *multiple critical factors*. We follow the standard approach [R5,R6] and consider six factors: driving smoothness, safety (collision or not), target reaching, velocity, orientation, and traveling time. The degradation of the  learning performance indicates that certain properties learned before is degraded, e.g., the safety and/or the driving smoothness is compromised. Furthermore, we evaluate the driving performance using standard metrics such as Time-to-Collision (TTC) and route completion rate (success rate), which are widely accepted in the autonomous driving community and is different from our training reward. In Tables 1-3, it can be seen that the driving performance degradation is indeed related to the training reward degradation.
>
> |    | Reward | Evaluation Metrics |
> | :--------: | :-------: | :-------:|
> | Choice  | driving related factors (ref. Appendix C )    | Success Rate and TTC |
> | Purpose |   Learning Signal for RL  | Driving Performance Evaluation|
>
> Moreover, we note that our reward design follows the standard approach [R5,R6] and incorporates several safeguards for robustness. First,  our reward function is designed based on fundamental physics-based metrics that remain meaningful across different scenarios and distributions. Second, we empirically validate this robustness through our experiments across diverse scenarios.
>
>
> - [R5] Knox, W. Bradley, et al. "Reward (mis) design for autonomous driving." Artificial Intelligence 316 (2023): 103829.
> - [R6] Kiran, B. Ravi, et al. "Deep reinforcement learning for autonomous driving: A survey." IEEE Transactions on Intelligent Transportation Systems 23.6 (2021): 4909-4926.
>
>
> **A2. (Parameters  $C\_1$ and $C\_2$)**  Parameter $C\_1$ and $C\_2$ are obtained by re-organizing the items on the RHS of the inequality in Theorem 1   and we add the detailed derivation Appendix B.
>
> **A3. (Switching strategy )** The purpose of the switching strategy is to address "when should we finetune the dynamics model and planning policy", where the priority of the finetuning is crucial for effective learning. It can be seen in Figure 1 that the finetuning strategies play a pivotal role in the learning process during online interaction. For instance, when model and policy are finetuned alternately (one model finetune followed by one step of policy finetune), the learning process can be hindered greatly since the major root cause for the performance degradation is not addressed effectively during finetuning (e.g., the model mismatch). To compare the different finetuning strategies impact on the leanring progress, we update the comparison Tables 1-3 in our revision.
>
>
> **A4. (Finetuning)** We thank the reviewer for pointing out this interesting direction. In response to the question in Weakness 3, we clarify that our current work focuses on demonstrating AdaWM's effectiveness on world model-based policies as they naturally align with our adaptation framework's design principles. VAD and UniAD follow fundamentally different architectural paradigms (end-to-end supervised learning), thus special design is needed to learn a world model that can represent the agent's understanding of the environment, which inevitably will change the VAD and UniAD structure.
>
> Moreover, our approach is general and model-agnostic, where the mathematical formulation and adaptation mechanism can theoretically be integrated with any world model-based driving system. The current evaluation focuses on demonstrating this fundamental effectiveness of real-time adaptation, which is a capability that supervised learning-based methods like VAD and UniAD inherently cannot achieve due to their *offline training nature*.
>
> We hope that we clarified these very helpful comments. We would be glad to engage in discussion if there are any other questions or parts that need further clarifications.

---

> > ### Comment · Reviewer_EsQt · 2024-11-24
> >
> > Most of my concerns have been addressed well by carefully reviewing the authors' thoughtful responses. I think the key merit of this article is the proposed idea of updating the world model and policy selectively and adaptively, which poses an interesting direction on how to optimize these two components more effectively.
> >
> > Regarding the evaluation protocols, I still suggest the authors report the driving score for comprehensive comparisons. Instead of simply ignoring this metric, authors should integrate your reply A1(3) into the main paper or supplementary to tell the readers why
> > the driving score might not be perfectly indicative of the policy performance. By doing so, the quality of this paper could be further strengthened.
> >
> > Overall, my final score for this paper would be 6 (marginally above the acceptance threshold).

---

### Meta-Review · Area_Chair_5GY5 · 2024-12-20

**Metareview:**

This paper presents a world model-based autonomous driving approach that specifically attempts correct for distribution shifts separately in the policy and dynamics model during fine-tuning. The paper evaluates the method on CARLA simulated driving tasks. The paper claims the method significantly improves the finetuning process. Theoretical analysis provided in the paper is used to motivate a component of the finetuning algorithm.

### Strengths
Reviewers mentioned the empirical results, that it was easy to grasp the main idea, that extensive experiments demonstrate the method's clear advantage, that the paper introduced a novel method to quantify the two mismatches that lead to performance degradation, and that the mathematical derivation is impressive.

### Weaknesses
Reviewers mentioned that there exist unverified model configurations and experimental settings, concerns about the absence of results on the Bench2Drive or CARLA Leaderboard-2 settings, potentially unfair comparisons to other baselines due to input mismatch, missing visualizations, missing comparisons to more model-based / world model-based planning methods, the potential  of requiring model and policy rollouts [during training], missing discussion / analysis of performance as a function of task complexity, unclear results of how LoRA-based finetuning affects performance, and uninvestigated performance in corner cases, lacking clarity in conciseness in the introduction, failing to adequately highlight the significance, and an inappropriate selection of baseline methods

Reviewers generally had their concerns addressed by the author responses. One reviewer was not, stating that the baselines may not be the most suitable, resulting in a lack of comprehensive experimental comparisons. Upon further inspection, I tend to agree that the baseline selection is underwhelming, but that the absence of these baselines does not undermine the paper's main claim that the finetuning method is effective. Reviewers and I both found that this claim was supported by evidence.

**Additional Comments On Reviewer Discussion:**

As mentioned above, reviewers generally had their concerns addressed by the author responses. One reviewer was not, stating that the baselines may not be the most suitable, resulting in a lack of comprehensive experimental comparisons.  Upon further inspection, I tend to agree that the baseline selection is underwhelming, but that the absence of these baselines does not undermine the paper's main claim that the finetuning method is effective. Reviewers and I both found that this claim was supported by evidence.

---

### Decision · Program_Chairs · 2025-01-22

Accept (Poster)